

# Linkages between land initialization of the NASA-Unified WRF v7 and biogenic isoprene emission estimates during the SEAC[4]RS and DISCOVER-AQ airborne campaigns

Min Huang[1,2], Gregory R. Carmichael[3], James H. Crawford[4], Armin Wisthaler[5,6], Xiwu Zhan[7],
Christopher R. Hain[2,a], Pius Lee[8], Alex B. Guenther[9]

[1]George Mason University, Fairfax, VA, USA
[2]University of Maryland, College Park, MD, USA
[3]University of Iowa, Iowa City, IA, USA
[4]NASA Langley Research Center, Hampton, VA, USA
[5]University of Oslo, Oslo, Norway
[6]University of Innsbruck, Innsbruck, Austria
[7]NOAA National Environmental Satellite, Data, and Information Service, College Park, MD, USA
[8]NOAA Air Resources Laboratory, College Park, MD, USA
[9]University of California, Irvine, CA, USA
[a]Now at: NASA Marshall Space Flight Center, Huntsville, AL, USA

*Correspondence to*: Min Huang (mhuang10@gmu.edu)

**Abstract.** Land and atmospheric initial conditions of the Weather Research and Forecasting (WRF) model are often interpolated from a different model output. We perform case studies during NASA's SEAC[4]RS and DISCOVER-AQ Houston airborne campaigns, demonstrating that initializing the Noah land surface model directly using a coarser resolution dataset North American Regional Reanalysis (NARR) led to significant positive biases in the coupled NASA-Unified WRF (NUWRF, version 7)'s (near-) surface air temperature and planetary boundary layer height (PBLH) around the Missouri Ozarks and Houston, Texas, as well as poorly partitioned latent and sensible heat fluxes. Replacing the land initial conditions with the output from a long-term offline Land Information System (LIS) simulation can effectively reduce the positive biases in NUWRF's surface air temperature fields by ~2°C. We also show that the LIS land initialization can modify the surface air temperature errors almost ten times as effectively as applying a different atmospheric initialization method. The LIS-NUWRF based isoprene emission calculations by the Model of Emissions of Gases and Aerosols from Nature (MEGAN, version 2.1) are at least 20% lower than those computed using the NARR-initialized NUWRF run, and are closer to the aircraft observation-derived emissions. Higher resolution MEGAN calculations are prone to amplified errors on small scales, possibly resulted from some limitations of MEGAN's parameterization and its inputs' uncertainty. This study emphasizes the importance of proper land initialization to the coupled atmospheric weather modeling and the follow-on emission modeling, which we anticipate to be also critical to accurately representing other processes included in air quality modeling and chemical data assimilation. Having more confidence in the weather inputs is also beneficial for determining and quantifying the other sources of uncertainties (e.g., parameterization, other input data) of the models that they drive.



## 1 Introduction

The weather-dependent emissions of biogenic volatile organic compounds (BVOCs), including the highly reactive species isoprene ($C_5H_8$), contribute to the formation of secondary short-lived climate pollutants such as ozone ($O_3$) and secondary organic aerosol (SOA). Therefore, these emissions affect air quality on local, regional, and global scales, which feed back to

the climate. For example, a modeling study by Li et al. (2007) showed that a 50% of reduction in Houston isoprene emissions led to 5-25 ppbv summertime afternoon $O_3$ reductions at its urban areas, and the transport of isoprene from north of the urban Houston area had non-negligible impact on its isoprene budget within several days. Wiedinmyer et al. (2005) found that summertime isoprene emitted from the Missouri Ozarks (also known as "isoprene volcano", where a high density of oak trees efficiently emit isoprene), along with its oxidation product formaldehyde (HCHO), can be transported eastward

and northward to urban areas (e.g., Chicago and St. Louis) to affect their $O_3$ burdens. Ozone and peroxyacetyl nitrate (PAN) produced from isoprene and other $O_3$ precursors can also affect air quality on hemispheric scale. Fiore et al. (2011) reported that for summer and fall, the $O_3$ sensitivity over Europe and North Africa to a 20% change in North American (NA) isoprene emissions was more than half of the sensitivity to a 20% change in NA anthropogenic emissions, greater than the $O_3$ response over the NA. Therefore, possible increases in future isoprene emissions due to the land cover and climate change

may offset the surface $O_3$ decreases due to controlling the anthropogenic emissions in NA and its downwind continents.

The Model of Emissions of Gases and Aerosols from Nature (MEGAN, Guenther et al., 2006, 2012) has been frequently used to generate BVOC emissions on flexible scales for air quality modeling. MEGAN computes emissions based on the emission source types and their densities, ambient carbon dioxide ($CO_2$) concentrations, and the meteorological conditions

(e.g., temperature, solar radiation, and moisture). It has been found that the MEGAN emissions are often higher than those calculated using other emission models, and are possibly associated with positive biases (e.g., Millet et al., 2008; Warneke et al., 2010; Carlton and Baker, 2011; Canty et al., 2015; Hogrefe et al., 2011). These biases, which still need careful validation with observation-based emission fluxes, can pose significant difficulties to accurately simulating isoprene and secondary air pollutants by chemical transport models. In addition to the impact of MEGAN parameterization, the outdated/unrealistic

land cover input data and the uncertainties of the meteorological inputs are important causes of these biases (e.g., Guenther et al., 2006, 2012; Carlton and Baker, 2011). The positive biases in the surface air temperature and radiation fields from meteorological models have been identified as major sources of uncertainty, and certain solutions have been established to reduce the biases such as substituting the modeled radiation with satellite radiation products. Much less has been done to explore the biases imported from the modeled soil moisture fields although satellite observations have suggested negative

correlations between BVOC emissions and soil moisture (e.g., Duncan et al., 2009). MEGAN emission sensitivity calculations by Sindelarova et al., (2014) showed weak direct impact of soil moisture on the isoprene emissions over vegetated and moist surfaces. However, the variability in soil moisture can indirectly impact BVOCs' emissions and their atmospheric distributions through affecting the air/canopy temperature and planetary boundary layer height (PBLH), the key





factors controlling the isoprene emissions and satellite column measurements (e.g., Palmer et al., 2003; Duncan et al., 2009), especially over the US transitional climate zones including the Great Plains and some east Asian regions (e.g., Miralles et al., 2012; Zeng et al., 2014; Lee et al., 2016; Zaitchik et al., 2012 and the references therein). Therefore, accurately simulating the land states and the correct representation of land-atmosphere interactions by meteorological models can

provide improved inputs for the MEGAN emission calculations.

The performance of coupled land-atmospheric modeling relies on numerous factors such as the choice of land surface model (LSM), nudging methods, and land use/land cover input data (e.g., Jin et al., 2010; Byun et al., 2011; Huang et al., 2016). The initialization of soil moisture and other land fields have also been shown important to the modeled atmospheric weather

states (e.g., air temperature, humidity, winds, precipitation, and the PBLH) and the latent/sensible heat fluxes. Suitable and sufficient LSM spin-up as well as land data assimilation can benefit the land surface modeling and the coupled atmospheric weather prediction (e.g., Rodell et al., 2005; Case et al., 2008; 2011; Zeng et al., 2014; Angevine et al., 2014; Collow et al., 2014; Lin and Cheng, 2015; Santanello et al., 2013, 2016). However, the potential benefit of appropriate land initialization of numerical weather models to emission and air quality related studies needs to be better understood.

In this study, we performed a number of NASA-Unified Weather Research and Forecasting (NUWRF) sensitivity simulations, in which different land and atmospheric initialization methods and model grid resolutions were experimented. The simulated weather states, especially the key variables impacting isoprene emissions, as well as heat fluxes were evaluated against in-situ and remote sensing observations. Isoprene emissions were then calculated by MEGAN, driven by

these various NUWRF simulations, and these estimates were compared with the aircraft observation-derived during two NASA airborne campaigns in September 2013. The paper is structured as follows: We will first introduce the isoprene emissions calculated by MEGAN (Section 2.1) and observations (Section 2.2), followed by the model evaluation datasets (Section 2.3). The NUWRF performance and its impact on isoprene emission estimates will be shown on a specific day in September 2013 (Section 3.1), as well as for multiple days in that month when research flights were executed for an airborne

campaign, which will then be further compared with the decadal mean conditions (Section 3.2). The sources of uncertainty of the emissions will be discussed in detail (Section 3.3) before the conclusions and suggestions on future directions are given in Section 4.

## 2 Methods and data

### 2.1 Bottom-up emission calculations

#### 2.1.1 MEGAN model version 2.1

The most recent version of MEGAN (version 2.1, Guenther et al., 2012) generates the net primary biogenic emissions that escape into the atmosphere, i.e., these are only emissions from the canopy to the atmosphere and do not include the chemical





fluxes from the atmosphere into the canopy, which on average can be a few percent of the net primary emissions (Guenther et al., 2012). The emissions are estimated based on Equation (1):

Emission = $[\varepsilon][\gamma][\rho]$ (1)

where $[\varepsilon]$ stands for the emission factor at standard conditions, $[\rho]$ accounts for the production and loss within the plant

canopies. $[\gamma]$ is a unitless emission activity factor, a product of multiple factors that account for the emission response to light ($\gamma_P$), temperature ($\gamma_T$), soil moisture ($\gamma_{SM}$), leaf age ($\gamma_A$), leaf area index (LAI), as well as $CO_2$ inhibition ($\gamma_{CO2}$), the process that reduces isoprene emissions when ambient $CO_2$ concentration increases above the level of 400 ppmv which needs to be better understood. Among the meteorological variables, MEGAN emissions are strongly sensitive to radiation and air temperature (Guenther et al., 2012, and the references therein), but less sensitive to soil moisture over vegetated and

moist surfaces including the central/southeastern US (Sindelarova et al., 2014), where the root zone soil moisture is usually larger than a threshold (the sum of a small empirical value and the soil type-dependent wilting point) above which $\gamma_{SM}=1.0$.

The stand-alone version of the MEGAN 2.1 was used in this study, which requires the users to provide the meteorological and land cover inputs. The land cover and meteorological inputs we used in this study will be introduced in detail in Sections 2.1.2 and 2.1.3, respectively. We ignored the $CO_2$ impacts on the emissions (i.e., $\gamma_{CO2} = 1.0$), as the monthly mean

$CO_2$ concentration in September 2013 was measured to be nearly 400 ppmv (i.e., 393.31 ppmv) at the Mauna Loa, Hawaii Observatory (https://weatherdem.wordpress.com/2013/10/10/september-2013-co2-concentrations-393-31ppm). The sensitivity calculation by Sindelarova et al. (2014) showed that for the year of 2003, the inclusion of $CO_2$ impact resulted in a 2.7% increase in MEGAN emissions globally under the 373.1237 ppmv $CO_2$ environment, corresponding to a $\gamma_{CO2}$ of 1.0277. Therefore, omitting the $CO_2$ impacts in this study would not introduce large biases. The $\gamma_{SM}$ value was also 1.0, as

the root zone soil moisture from our meteorological input exceeded the sum of the empirical value and the wilting point (from Chen and Dudhia, 2001) over the regions of interest.

### 2.1.2 Plant functional type (PFT) and LAI input data

The recommended high-resolution 30 arc-second PFT input files for the year of 2008 (Guenther et al., 2012; http://lar.wsu.edu/megan/docs/NorthAmericaPlantFunction/), based on the Community Land Model 16 PFT classification

system, were interpolated to the NUWRF model grids for use in this study. The LAI input was based on the Terra-Moderate Resolution Imaging Spectroradiometer (MODIS) 8-day product, and the grids with missing data were filled with the monthly-mean MODIS product. Figure 2d shows the LAI input for MEGAN on 11 September, 2013 over the Greater Houston area, indicating denser vegetation north and northeast to the downtown Houston area.





### 2.1.3 The NASA-Unified WRF meteorological simulations using different land and atmospheric initialization methods

The MEGAN emission calculations in this study were driven by the meteorological fields simulated by the NUWRF (Peters-Lidard et al., 2015) modeling system version 7. The WRF component within this version of NUWRF was modified from the

core WRF version 3.5.1, and it simulates atmospheric processes on a terrain-following mass vertical coordinate system over flexible spatial and temporal scales (Skamarock et al., 2008). The NUWRF supports coupling between WRF and NASA's Land Information System (LIS), a flexible land surface modeling/assimilation framework developed to integrate satellite and ground observations and advanced land surface modeling techniques to produce optimal fields of land surface states and fluxes (Kumar et al., 2006, 2008). This coupled system enables the investigation of land-air interactions including evaluating

the impact of land initialization and land data assimilation on the atmospheric states (e.g., Santanello et al., 2016).

A number of NUWRF meteorological simulations (Table 1) were performed over the contiguous US (CONUS) for September 2013 on 12 km (479×399 grids) and 4 km (1248×900 grids) horizontal resolution Lambert conformal grids that are both centered at 40°N/95°W. Same as in Huang et al. (2016), the vertical grid spacing recommended by the Texas Commission on Environmental Quality (TCEQ) was implemented. The four-soil layer (with thicknesses of 0.1, 0.3, 0.6, 1.0

m) Noah LSM (Chen and Dudhia, 2001), an option widely used in scientific and operational applications, was applied. The Noah LSM is based on grid-dominant land use/land cover types and we chose to use the recommended International Geosphere-Biosphere Programme (IGBP)-modified MODIS 20-category land use/land cover, which reflect more recent conditions than the other available options (Tao et al., 2013; Yu et al., 2012). The commonly-used Mellor-Yamada-Janjic PBL scheme (Janjic, 2002) and the matching Monin-Obukhov (Janjic Eta) surface layer scheme (Monin and Obukhov,

1954) were chosen, although these might lead to shallower, cooler PBL and less vertical mixing than other available schemes in WRF (e.g., Saide et al., 2011; Angevine et al., 2012; Huang et al., 2013; Zhang et al., 2016). Other key physics options include: the Eta microphysics (Rogers et al., 2001), the Rapid Radiative Transfer Model (RRTM) short- and long-wave radiation (Iacono et al., 2008), and the Betts-Miller-Janjic cumulus parameterization (Janjic, 2000). The urban surface option (Chen et al., 2011) was turned on only in the 4 km simulations. These simulations were started at 06 UTC (00 Central

US Standard Time) of each day. The 4 km and 12 km calculations used 4s and 24s time steps, respectively, and they were recorded hourly at full clocks for 24 and 48 hours, respectively. The effect of simulation length (i.e., day 1 and day 2-forecasts, defined as the simulations 00-24h and 25-48h since the initial time, respectively) on the 12 km NUWRF performance will be included in the discussion.

Our NUWRF simulations include three sensitivity simulations that evaluate the impact of two land initialization methods, as

illustrated in Figure 1: a) A usual method applied to the 12 km NUWRF grid, in which the atmospheric and land initial conditions (IC) were downscaled from the output of a coarse model North American Regional Reanalysis (NARR, at 32 km horizontal resolution with a 3-hourly time interval, Mesinger et al., 2006). NUWRF's atmospheric lateral boundary





conditions (LBC) were also downscaled from NARR. NARR is known to be generally drier and warmer than the observations (e.g., Royer and Poirier, 2010; Kennedy et al., 2011). Same as in default and many WRF simulations, the green vegetation fraction (GVF) input data in this case were based on climatological monthly mean satellite normalized difference vegetation index (NDVI). Although realistic vegetation density in land surface models is important to accurately

representing the partitioning of soil evaporation and canopy transpiration (e.g., Bell et al., 2012), the model did not show considerable sensitivities in response to replacing the climatological monthly GVF with the satellite near real-time GVF over this study's focused regions, and these will be briefly discussed in Section 3.1.1; b) The 12 km and 4 km "control (ctrl) simulations": Same as a), except that NUWRF's land IC were instead from the output of long-term (i.e., cold-started from 01 January, 2001, cycled twenty times from 01 January, 2001 to 31 December, 2001 before running all the way through

September 2013) offline LIS simulation that allowed the land conditions to reach thermodynamical/water equilibrium. The LIS offline spin-up was completed on the same horizontal resolutions as NUWRF, forced by highly resolved atmospheric fields from the Global Data Assimilation System (GDAS) and precipitation data from the Global Land Data Assimilation System (GLDAS). The daily near real-time satellite GVF were used within LIS and NUWRF. Two additional NUWRF "control" simulations were conducted at 12 km and 4 km resolutions to assess the impact of atmospheric IC and LBC on

NUWRF's performance. NUWRF's IC and LBC in these simulations were taken from the atmospheric fields of the North American Mesoscale Forecast System (NAM, at 12 km horizontal resolution with a 6-hourly time interval, Janjic, 2003; Janjic et al., 2004), which is known to usually have positive biases in temperature, moisture, and wind speed in the CONUS (e.g., McQueen et al., 2015a, b).

## 2.2 Emissions derived from in-situ isoprene measurements

The mixed-PBL approach introduced by Warneke et al. (2010) was adopted to derive the isoprene emissions during two NASA airborne campaigns in September 2013, which were compared with the NUWRF-MEGAN bottom-up emissions. The mixed-PBL approach accounts for isoprene's atmospheric lifetime but neglects the impact of horizontal advection, and it estimates isoprene emissions based on Equation (2):

$$\text{Emission}_{\text{isoprene}} - F_e = [\text{isoprene}] \times \text{boundary layer height} \times k_{OH} \times [OH] \tag{2}$$

where [isoprene] and [OH] are the concentrations of isoprene and hydroxyl radical (OH), respectively, and the data used in our calculations will be introduced in Sections 2.2.1-2.2.2; $k_{OH}$ refers to the rate coefficients with OH which was set to be $101 \times 10^{12}$ cm$^3$/molecule/s; and $F_e$ represents the entrainment flux from the boundary layer to the free troposphere, set constantly to be 30% of the emission flux, based on aircraft isoprene flux measurements over the Amazonian rain forest (Karl et al., 2007). Our NUWRF-modeled PBLHs, after qualitatively evaluated with the aircraft measurements, were used in

the emission calculations. The uncertainty of the isoprene emissions derived by this approach will be further discussed in Section 3.3.



### 2.2.1 Isoprene measurements

NASA's Studies of Emissions and Atmospheric Composition, Clouds and Climate Coupling by Regional Surveys (SEAC[4]RS, https://espo.nasa.gov/home/seac4rs/content/SEAC4RS) was conducted in August-September 2013, when more than 20 research flights were executed. The August deployment over the western US focused on the influences of biomass

burning pollution, their temporal evolution, and their impacts on meteorological processes and air quality. The September deployment over the southeastern US primarily focused on the attribution and the quantification of trace gas pollution and their distributions as a result of deep convection. In-situ isoprene data measured by the Proton Transfer Reaction-Mass Spectrometry (PTR-MS, de Gouw and Warneke, 2007) on board the DC-8 aircraft were used in the emission calculations. We used data obtained in the Missouri Ozarks ("isoprene volcano") region where the biogenic isoprene emissions were high

and the potential measurement interferences from furan and 2,3,2-methylbutenol (232-MBO) were negligible: Furan is found in significant concentrations only in biomass burning plumes and no enhancement in the biomass burning tracer acetonitrile was observed in our case studies (details in Section 3.1). 232-MBO is only emitted from the coniferous ecosystems in the western US. The isoprene observation data have an accuracy of ±5%.

Nine research flights were executed in September 2013 (i.e., on 4, 6, 11, 12, 13, 14, 24, 25, 26 of this month) to support the Houston portion of the NASA DISCOVER-AQ (Deriving Information on Surface conditions from Column and Vertically Resolved Observations Relevant to Air Quality, Crawford et al., 2014, https://discover-aq.larc.nasa.gov) field experiment. High-resolution in-situ isoprene measurements during the DISCOVER-AQ were made using a proton-transfer-reaction time-of-flight mass spectrometry instrument (PTR-ToF-MS, with an accuracy of ±10%, Müller et al., 2014) on board the P-3B

aircraft over selected locations in the Greater Houston area three times a day (i.e., morning, noon-early afternoon, and mid-afternoon) to explore their spatial and diurnal variability. The isoprene measurements from the PTR-ToF-MS are possibly interfered by other VOCs from anthropogenic sources in Houston (e.g., from the oil and gas industries). Therefore, we focus on deriving biogenic emissions at the Conroe site, a region north to the downtown Houston area with medium vegetation coverage (Figure 2d) and less strongly influenced by urban transportation/industrial sources and biomass burning plumes, as

indicated by the isoprene-carbon monoxide (CO) relationships in Figure A1. Additionally, we investigated the hourly surface isoprene measurements available at eight TCEQ Automated Gas Chromatograph (AutoGC) monitoring stations, mostly located in the downtown Houston area (marked as triangles in Figure 2d). The data before the sunrise and after the sunset, when biogenic emissions are at their daily minima, are particularly useful for determining the regional background and non-biogenic isoprene levels, and therefore they helped quantify the uncertainty in the observation-derived emissions.

The Limit of Detection (LOD) applied to all AutoGC target compounds is currently 0.4 ppbC (0.08 ppbv for isoprene).

The ground speed of the DC-8 and P-3B aircraft was around 8-9 km/minute near the "isoprene volcano" areas during SEAC[4]RS and 9-14 km/minute at around Conroe during DISCOVER-AQ within the focused time period. Therefore, the aircraft data averaged in 1-minute interval (released on 10 February, 2016 and 23 July, 2015 for SEAC[4]RS and DISCOVER-



AQ, respectively) were used to estimate the emissions, as they represent isoprene concentrations on similar spatial scales to the NUWRF-MEGAN's. At around Conroe, multiple P-3B aircraft data points correspond to several NUWRF model grids, and the averaged emissions based on NUWRF-MEGAN and the median PBL observations were used in the comparisons.

### 2.2.2 OH from the NOAA National Air Quality Forecasting Capability (NAQFC)

Due to the lack of aircraft OH measurements in September 2013, the OH concentrations simulated by the NOAA NAQFC 12 km Community Modeling and Analysis System (CMAQ, Byun and Schere, 2006; Pan et al., 2014) were used to derive isoprene emissions. The NAQFC CMAQ is driven by the NAM meteorological fields, and the biogenic emissions are computed online from the Biogenic Emissions Inventory System (BEIS) version 3.14, that often produces much lower emissions than MEGAN at the "isoprene volcano" and in the eastern Texas (e.g., Warneke et al., 2010; Carlton and Baker, 2011). The NAQFC CMAQ OH performance near the "isoprene volcano" was generally satisfactory for the studied period: i.e., the mean±standard deviation of the predicted OH of $(1.8\pm0.8)\times10^6$ on 11 September and $(1.5\pm0.3)\times10^6$ molecule/cm$^3$ on 06 September along the Missouri flight paths (to be shown in Sections 3.1 and 3.3) are of the close magnitudes to the observationally constrained OH concentrations in that area during SEAC$^4$RS (e.g., $(1.3\pm0.3)\times10^6$ molecule/cm$^3$ by Wolfe et al., 2015). Close to the estimated OH concentrations near Houston on 16 September, 2006 (Warneke et al., 2010), the simulated PBL OH on 11 September, 2013 range from $\sim1.7\times10^6$ to $\sim4.0\times10^6$ molecule/cm$^3$ along the P-3B around Conroe, and the averaged OH levels on all P-3B flight days are also within this range. Little prior knowledge exists on the CMAQ OH performance in the Greater Houston area, except the moderate negative biases (with observed-to-modeled ratios of 1.15-1.36) reported by Czader et al. (2013) for May 2009. As their modeling system was configured differently from the NAQFC, the biases of the modeled OH fields from the NAQFC CMAQ system need to be investigated further in the future.

### 2.3 Evaluation datasets

### 2.3.1 Ground and aircraft measurements of air temperature, solar radiation, and PBLH

We focus on evaluating the sensitivities of NUWRF's air temperature, solar radiation, and PBLH to the initialization methods, as they are the most important weather variables to the estimated isoprene emissions. The NUWRF-modeled (near-) surface air temperature fields were compared with the National Center for Environmental Prediction (NCEP) Global Surface Observational Weather Data (also used in Huang et al., 2016), the DC-8 aircraft air temperature measurements, and the 5-minute TCEQ special observations at the Conroe site taken in support of the airborne campaigns. The NUWRF-modeled solar radiation were briefly compared with the measurements by pyranometers on board the DC-8 and at Conroe. The NUWRF-simulated PBLH were also roughly compared with the estimated PBLH by: a) the Differential Absorption Lidar (DIAL)-High Spectral Resolution Lidar (HSRL) measurements on board the DC-8 aircraft during the SEAC$^4$RS





campaign (Figure 2a); and b) the vertical gradients of the in-situ isoprene observations measured on board the P-3B aircraft during DISCOVER-AQ at around the Conroe site at different times of the day (Figure 2c).

### 2.3.2 Satellite soil moisture and heat flux products

The European Space Agency (ESA) soil moisture Climate Change Initiative (CCI, http://www.esa-soilmoisture-cci.org)
project produces daily surface soil moisture data during 1978-2014 at 0.25°×0.25° horizontal resolution, based on multiple passive and active sensors, as well as by merging both passive and active products. Long-term soil moisture changes in the US based on the CCI product contributed to the US National Climate Assessment report (Melillo et al., 2014, p72-73). Fang et al. (2016) reported that the merged CCI product exhibited higher anomaly correlation (than the individual active/passive products) with both Noah LSM simulations and in-situ measurements during 2000-2013. Therefore, the most recent version
of this merged product (version 02.2, released in 2015) with enhanced spatial and temporal coverage and intercalibration between different instruments was used to evaluate the modeled soil moisture fields and the normalized soil moisture anomalies (as defined in Equation (3)), at where the CCI data quality flag equals zero:

$$\text{Normalized soil moisture (SM) anomaly} = \frac{daily\ SM - monthly\ mean\ SM}{monthly\ SM\ standard\ deviation} \qquad (3)$$

Soil moisture controls the partitioning of energy into latent (the energy related to changes in phase) and sensible heat (the
energy related to temperature changes) fluxes. To evaluate the appropriateness of NUWRF's land initialization, we compared the NUWRF-modeled absolute heat fluxes and their partitioning (i.e., the evaporative ratio, defined as latent heat/(latent heat+sensible heat)) with the Atmosphere-Land Exchange Inversion (ALEXI, Anderson et al., 2007; Hain et al., 2011) retrievals. The ALEXI heat flux product using the NOAA Geostationary Operational Environmental Satellite (GOES) thermal-IR (TIR) land surface temperature, along with its soil moisture proxy retrievals, is a part of the NOAA operational
GOES Evapotranspiration and Drought Product System (GET-D; http://www.ospo.noaa.gov/Products/land/getd). Although limited to clear-sky conditions, ALEXI provides retrievals over a wide range of vegetation cover on horizontal resolution close to our NUWRF's (i.e., 0.08°×0.08° for this study).

## 3 Results and discussion

### 3.1 Case study of 11 September, 2013

We first show a case study on a specific day of 11 September, 2013, when aircraft measurements were available from both the SEAC[4]RS and DISCOVER-AQ campaigns. For SEAC[4]RS, the DC-8 aircraft sampled over broad areas in the central/southeastern US on this day (Figure 2a), passing the "isoprene volcano" region in Missouri at the early afternoon time (18:30-19:30 UTC/12:30-13:30 local standard time), where mixed layer heights indicated by the DIAL-HSRL



instrument were mostly below 2 km. Elevated isoprene concentrations (up to ~10.4 ppbv) were observed by the PTR-MS near the surface (<1 km, a.g.l.). Biomass burning plumes had little interference to these isoprene measurements, as determined by the low acetonitrile concentrations (Figure 2b) in the sampled airmasses. For DISCOVER-AQ, the P-3B repeatedly took measurements at different times of the day at around the Conroe site in Houston (Figure 2d), where the

observed isoprene vertical profiles indicate the growth of PBLH from the morning (a few hundred meters, a.g.l.) to the afternoon (~2 km, a.g.l.), and ~50% higher near-surface isoprene concentrations in the afternoon (~2.5 ppbv) than in the morning (~1.7 ppbv) (Figure 2c). The CO concentrations in the sampled airmasses were below 200 ppbv (Figure A1), indicating negligible biomass burning source impacts. Anthropogenic emission sources are mainly located at the downtown Houston, where the daytime P-3B aircraft isoprene concentrations (i.e., at the Moody Tower, Deer Park, Channelview

spirals) did not exceed ~0.6 ppbv and presented a different isoprene-CO enhancement ratio from Conroe's (Figure A1). The magnitudes of the downtown aircraft isoprene measurements were slightly lower than most of the nearby surface measurements during the daytime (Figure 2e) that were ~twice as high as their isoprene levels during the nighttime (~0.2-0.3 ppbv) when biogenic emissions are at their daily minima. Therefore, we expect that the non-biogenic emissions contributed to no more than 0.3 ppbv of the P-3B observed isoprene over that region.

**3.1.1 Evaluation of NUWRF surface air temperature, PBLH, soil moisture, and heat fluxes**

Figure 3a compares the NUWRF modeled surface air temperature in the central/eastern US with the ground observations in the early afternoon where the DC-8 flew past Missouri. The 12 km usual run shows 2-4 °C positive biases in Missouri, which are of the similar magnitudes to the findings by Carlton and Baker (2011) for these regions. These positive biases were dramatically reduced in the NUWRF ctrl runs. Evaluation of the modeled air temperature and PBLH along the DC-8

flight paths in Missouri was summarized in Table 2. RMSEs of the modeled air temperature from the ctrl runs are ~1.5 °C lower than the 12 km usual run-produced, corresponding to thinner (~0.6 km on average) and less spatially variable PBLH which may be closer to the reality referring to the DIAL-HSRL data (that can also be quite uncertain). The higher resolution 4 km ctrl run generated slightly (~0.04 °C) better air temperature and ~0.02 km thinner mean PBLH than the 12 km ctrl run.

Figure 4a compares the NUWRF modeled daytime surface air temperature at the Conroe site against the TCEQ special

measurements, and Table 3 summarizes the statistical evaluation of the NUWRF PBLH and surface air temperature performance in Conroe. Similar to the conditions in Missouri, temperatures from the 12 km usual run are positively biased by 1.8-3.0°C during the daytime, and the ctrl runs significantly better captured the observed magnitudes (i.e., with 1.6-1.8 °C lower RMSEs than the 12 km usual run), corresponding to at least ~300 m lower PBLH, which are likely more realistic referring to the observed isoprene vertical profiles. The 4 km ctrl simulation produced noticeably lower air temperature and

PBLH at the morning (by up to ~0.5 °C/~170 m) and afternoon (by up to ~2.6 °C/~290 m) times than the 12 km ctrl run.




Figure 3b-c compare the CCI daily surface soil moisture fields with the NARR and LIS modeled at the NUWRF initialization time on this day. The NARR soil moisture fields are at least 0.1 m³/m³ drier than the LIS-NUWRF systems at the beginning of the simulation (Figure 3b), causing the spurious NUWRF temperature/PBLH fields as described earlier. The impact of initial soil moisture states on simulated temperature at later times is similar to the results in Collow et al.

(2014) for the Great Plains in May 2010. Figure 3c shows that the normalized soil moisture anomalies from NARR and LIS overall demonstrate similar spatial patterns, which was difficult to be validated with the CCI product due to too many missing data in September 2013. This suggests that when downscaling land fields to a different modeling system, adjusting the large-scale dataset based on the climatology (preferably for a much longer record) of both systems would be helpful. This adjustment, sometimes also called "bias-correction", is indeed useful in satellite land data assimilation (e.g., scaling

satellite soil moisture fields based on the model's and the satellite's climatology before they are assimilated). Figure 5 compares the modeled heat fluxes with the ALEXI retrievals, indicating that the usual land initialization method resulted in significantly underpredicted latent heat and overpredicted sensible heat, and the partitioning between these heat fluxes were poorly represented. Such evaluation confirmed that the usual land initialization method is inappropriate for this case.

It's worth pointing out that by replacing the WRF-default monthly-mean climatological GVF input with the daily near real-

time GVF in the 12 km usual run, we did not find significantly changes in the modeled temperature (i.e., <±0.5°C, as shown in Figure A2, right) and PBLH (not shown in figures) fields near the Missouri Ozarks and Conroe, where the GVF differences are within ±0.1 (Figure A2, left). Therefore, soil states at the initialization were the major causes to the different temperature and PBLH fields from the usual and ctrl runs over these regions. In contrast, weather fields over some other central/southeastern US regions, particularly in the eastern Arkansas, are shown very sensitive to this GVF update, with

negative (positive) GVF differences resulting in positive (negative) temperature differences. Over these regions, the different weather fields in usual and ctrl runs indicate the net effect of GVF and soil initialization.

Figure 4b evaluates the impact of simulation length on the modeled surface air temperature at Conroe, and in this case higher temperature biases were shown in the longer simulation (the day 2-forecast) regardless of the land initialization method, especially during the morning and early afternoon times. The RMSEs of daytime air temperatures from the day 2-

forecasts are ~0.3 °C higher than the day 1-forecasts. Figure 4c shows the impact of atmospheric IC/LBC on the modeled air temperature. In both 12 km and 4 km grids, replacing the NARR IC/LBC with NAM's resulted in larger temperature amplitude, associated with greater negative biases in the morning and positive biases at around the mid-afternoon. The RMSEs of daytime air temperatures from the NAM-related cases are ~0.2 °C higher than the NARR-related cases. Figures 4a and 4c together also suggest that an inappropriate land initialization for a regional simulation can result in almost ten

times larger model errors than using an alternative atmospheric IC/LBC.

The solar radiation fields from the multiple NUWRF runs were briefly evaluated. It was found that the regional NUWRF solar radiation fields in Missouri from the various runs are vastly similar in the early afternoon local time, >30% (a couple of





hundred of W/m$^2$) larger than the DC-8 measurements. These biases are close to what has been reported by Carlton and Baker (2011), and the WRF-satellite differences in Guenther et al. (2012). The daytime NUWRF solar radiations at Conroe had time-varying biases but on average are a few percent different from the observations, and the photosynthetically activate radiation at Conroe differed by up to ~12W/m$^2$ among these simulations.

**3.1.2 NUWRF-MEGAN and observation-derived isoprene emissions in Missouri and Houston**

Figure 6a shows the spatial distributions of the MEGAN isoprene emissions driven by the different NUWRF simulations, compared with the observation-derived emissions at the early afternoon time, when the DC-8 aircraft sampled at the "isoprene volcano" and the isoprene emissions approached the daily maxima. Similar spatial patterns of the MEGAN emissions were produced when different NUWRF runs were used. The emissions based on the 12 km NUWRF usual run are

at least 20% larger than those driven by the NUWRF ctrl runs, corresponding to a ~2 °C larger positive bias in NUWRF temperature. Such emission sensitivities to the air temperature are close to the magnitudes reported in literature for other regions (Guenther et al., 2006, 2012; Wang et al., 2011). The NUWRF-MEGAN calculated emissions are 22-49% higher than the observation-derived emissions along the DC-8 flight tracks, with the 4 km NUWRF ctrl run-based MEGAN emissions the closet to the observation-derived.

Figure 6b shows the spatial distributions of the MEGAN isoprene emissions driven by the different NUWRF runs over Houston near the local standard noon time, the second time P-3B sampled over Conroe on that day, when isoprene emissions almost reached the daily maximum. Similar to the Missouri conditions, the MEGAN emissions driven by the 12 km NUWRF usual run are >20% larger than the cases driven by the NUWRF ctrl runs. Figure 7a compares the NUWRF-MEGAN daytime isoprene emissions at Conroe. The 12 km NUWRF usual run-based daily peak emissions during local

noon/early afternoon times are ~20% higher than the 12 km NUWRF ctrl run-based, the latter of which is closer to the observation-derived. The daytime-integrated emissions derived using the 12 km NUWRF usual run are ~21% higher than the 12 km NUWRF ctrl run-based. Again this discrepancy corresponds to a ~2 °C temperature differences on this day (Figure 4a; Table 3). The emissions driven by the 4 km NUWRF ctrl run are the lowest, with the daytime integrated and the peak emissions ~40% lower than the 12 km NUWRF ctrl run-based, and they substantially deviate from the observation-

derived. This is in part due to the coolest temperature from this NUWRF run, especially in the afternoon, as well as its weakest photosynthetically activate radiation than the 12 km simulations' (i.e., by ~10 W/m$^2$ on average during the daytime). Uncertainty may also be resulted from some limitations of MEGAN's parameterization and its other inputs (e.g., PFT and LAI) on grid scale.

The impacts of simulation length and atmospheric initialization on the NUWRF-MEGAN isoprene emissions at Conroe are

generally much smaller than the impact of land initialization (Figures 7b-c), mainly due to the smaller temperature sensitivities (Figures 4b-c): The day-2 forecast derived emissions are higher than the day 1 forecast-based emissions by



~10% in Conroe at the local standard noontime, but their daytime-integrated isoprene emissions differ much less (~1.5%). Daytime maximum emissions disagree by only <±2% in both grids. The noontime isoprene emissions related to NAM and NARR IC/LBC differ by less than 2% in both resolutions, and the daytime-integrated emissions related to NAM IC/LBC are higher than the NARR related by 0.8% and 5.2% in 12 km and 4 km grids, respectively.

## 3.2 Conditions on extended time periods

### 3.2.1 Conditions on multiple flight days during DISCOVER-AQ in September 2013

As the 12 km NUWRF ctrl run-based MEGAN isoprene emissions showed the best agreement with the observation-derived emissions at Conroe on 11 September (Section 3.1), we calculated MEGAN isoprene emissions using this set of NUWRF simulation also for the other eight DISCOVER-AQ flight days when variable meteorological conditions were present (details are in the flight reports at: https://discover-aq.larc.nasa.gov/planning-reports_TX2013.php), and the multi-flight day averaged MEGAN calculations were compared with the P-3B aircraft observation-derived at the Conroe site (Figure 8). The multi-day averaged MEGAN and observation-based emissions are higher than the estimates for 11 September, except in the morning. The multi-day mean morning emissions from MEGAN are ~44% higher than the observation-derived, a larger discrepancy than on 11 September. A possible reason for this morning-time overestimation is that MEGAN does not account for the circadian control that can lower the isoprene emissions from some canopies (Hewitt et al., 2011). At local noontime and afternoon times, unlike the 11 September condition, the multi-day averaged MEGAN emissions were slightly (by <5%) lower than the observation-derived.

### 3.2.2 September 2013 comparing with decadal mean conditions

We extend the analyses to the interannual variability of drought and vegetation conditions in relation to the isoprene emissions in Conroe. The monthly anomalies were calculated for HCHO column (which is often used to derive biogenic emissions) from the Ozone Monitoring Instrument (OMI, De Smedt et al., 2015), Terra-MODIS LAI, ESA CCI microwave soil moisture and ALEXI TIR soil moisture proxy in September 2013, related to the decadal (2005-2014) September means. The eastern Texas was under extreme drought conditions in September 2011 as indicated by the Palmer Drought Severity Index (http://www.ncdc.noaa.gov/temp-and-precip/drought/historical-palmers.php), which was excluded from the decadal mean calculations as severe drought can reduce or terminate isoprene emissions (Pegoraro et al., 2004) and complicate the anomalies. At Conroe, close-to-1 anomalies are found in September 2013 for the ALEXI and CCI data (~0.99 and ~0.98), and vegetation was slightly less dense than the decadal mean conditions (the LAI anomaly of ~0.96). A much lower than average HCHO column (the anomaly of ~0.77) was observed by OMI in this month. A higher OMI anomaly (~0.99) was found in September 2006 studied by Warneke et al. (2010), under drier conditions (ALEXI and CCI anomalies of ~0.77 and ~0.91, respectively) with higher-than-average vegetation coverage (the LAI anomaly of ~1.07). Note that these interannual





differences can be complicated by the uncertainty in the satellite data, and also reflects the possible influences by the temporal changes in non-biogenic VOC emissions, the local/regional chemistry, and the plant types in this area.

### 3.3 Uncertainty discussions

In addition to the biases in NUWRF's surface air temperature, a number of other factors can affect the NUWRF-MEGAN isoprene emission calculations. These include:

a)  The outdated PFT data that represent the year 2008 conditions and the uncertainty in the MODIS LAI input.

b)  The known positive biases in NUWRF's solar radiation fields partially due to the lack of aerosol impacts and the misplaced/missing clouds. It has been shown that implementing certain satellite solar radiation product can reduce the

biases in MEGAN emissions for other time periods (Carlton and Baker, 2011; Guenther et al., 2012). Identifying suitable satellite radiation products for this case will be included in future work.

c)  As described in Section 2.1, the MEGAN version 2.1 net primary emissions are always larger than the net emission flux (by a few percent on average but may be larger at a specific location) due to the omission of deposition. Adding that contribution in future emission calculations is important.

d)  Other limitations in MEGAN's parameterization which good input data can help better diagnose.

The uncertainties of aircraft observation-derived isoprene emissions are expected to come from:

a)  The PTR-MS and PTR-ToF-MS measurements have accuracies of ±5% and ±10%, respectively, that can be propagated to the emission calculations. These were smaller than the ±15% from the Warneke et al. (2010) study.

b)  The biases introduced from the NAQFC CMAQ OH fields (as mentioned in Section 2.2.2), which will need to be investigated further on grid-scale (e.g., by comparing them with other modeling products covering our studied period).

c)  As discussed in Warneke et al. (2010), the mixed-PBL approach neglects horizontal transport, which may attribute transported isoprene to the incorrect grid boxes. For this study, the observed wind speed along the SEAC[4]RS DC-8 flight path ranged from 0.27 to 5.47 m/s, with the mean value of ~1.68 m/s. The TCEQ 5-minute surface wind speed

observations were no larger than <3.5 m/s on 11 September. Assuming isoprene lifetime in this study is ~an hour, the aircraft observed isoprene may be actually emitted from the nearby 1-2 model grids on the 12 km scale. Therefore, this approach introduces a random error which may not significantly affect the magnitude of regional emission calculations in Missouri but may have a larger impact on the Conroe case especially on a single day. Developing and applying top-down methods that also account for the atmospheric transport should be strongly encouraged.

d)  As discussed in Warneke et al. (2010), the constant 30% entrainment flux may not be realistic for the regions/times we studied, which needs further validation.

e)  Regional non-biogenic emission sources may contribute to 10-20% of the aircraft observed isoprene at Conroe, as estimated by the ground in-situ data (Section 2.2.1).



f) The mixed-PBL approach assumes complete vertical mixing which may not be true in practice. Additionally, the control run based NUWRF modeled PBLHs (Tables 2-3) were used, possibly associated with uncertainty on a magnitude of a few hundred meters (~20%).

Warneke et al. (2010) estimated the uncertainty of their aircraft observation-derived emissions over Texas to be a factor of 2 (-50%, +100%). We anticipate the uncertainty of ours to be of the similar magnitude for the single-day Conroe case, but smaller in the multi-day averaged emissions in Conroe. The regional-averaged aircraft observation-derived emissions over the "isoprene volcano" region (Figure 6a for 11 September, and Figure A3 for 06 September with more descriptions in the figure caption) from this study are close to the result in Wolfe et al. (2015) of $8\pm1$ mg/m$^2$/h, derived using a different method
for the similar regions during SEAC$^4$RS.

## 4 Conclusions and suggestions on future direction

We performed case studies during the SEAC$^4$RS and DISCOVER-AQ Houston field campaigns, showing that a usual method to initialize the Noah LSM (i.e., directly downscaling the land fields from the coarser resolution NARR) led to significant positive biases in the coupled NUWRF (near-) surface air temperature and PBLH around the Missouri Ozarks
and Houston, Texas, as well as poorly partitioned latent and sensible heat fluxes. Replacing the land initial conditions with the output from a long-term offline LIS simulation effectively reduced the positive biases in NUWRF's surface air temperature fields. We also showed that using proper land initialization modified NUWRF's surface air temperature errors almost ten times as effectively as applying a different atmospheric initialization method. The LIS-NUWRF based MEGAN version 2.1 isoprene emission calculations were at least 20% lower than those computed from the NARR-initialized
NUWRF run, closer to the aircraft observation-derived emissions. The higher resolution NUWRF-MEGAN calculations were prone to amplified errors on small scales, possibly resulted from some limitations of its parameterization and the inputs' uncertainty. This study emphasizes the importance of proper land initialization to the coupled atmospheric weather modeling and the follow-on biogenic emission modeling. We anticipate that the improved weather fields via improved land initialization will also benefit the representation of the other processes (other weather-dependent emission calculations,
transport, transformation, deposition) included in air quality modeling, and therefore can help improve the simulated chemical fields. The study is limited to selected locations and times considering the availability of aircraft data, and the observation-derived emissions may also be associated with large uncertainty. In future, developing methods to combine satellite land and atmospheric chemical data assimilation should be encouraged to further improve air quality modeling and top-down emission estimates over broader regions/extended time periods to help interpret the trends and variability of the
atmospheric composition. The improved chemistry output from regional models can also help evaluate the current "a priori" used in satellite retrievals, and may serve as an alternative.



It should be noted that many published model comparison studies did not adequately assess the impacts of model inputs versus their parameterization. Having more confidence in the weather inputs is beneficial for quantifying the other sources of uncertainties (e.g., parameterization, other input data) of the models that they drive. In future, the impact of atmospheric weather input on emissions computed using other biogenic emission models (e.g., BEIS, future versions of MEGAN) will be

explored, and efforts will be made to improve the other inputs data (e.g., radiation, land cover).

Although we recommend initializing WRF or NUWRF with the LIS land fields, when long-term atmospheric forcing data are not available to facilitate the offline LIS spin-up, we suggest adopting the self spin-up method (Angivine et al., 2014) or "bias-correcting" the initial condition model's land fields based on the climatology of the initial condition model and the

target model. Experimenting simulations with different LSMs along with suitable nudging methods can also be helpful.

## 5 Model and data availability

Instructions for obtaining and running the used models can be found at: LIS (lis.gsfc.nasa.gov/documentation/lis); NUWRF (nuwrf.gsfc.nasa.gov/doc); MEGAN (lar.wsu.edu/megan/docs). The satellite land products, and the NUWRF/MEGAN output can be made available upon request. The open access to the used aircraft and ground observations is acknowledged:

Aircraft data were obtained from: http://www-air.larc.nasa.gov/index.html

SEAC[4]RS: doi: 10.5067/Aircraft/SEAC4RS/Aerosol-TraceGas-Cloud

DISCOVER-AQ: doi: 10.5067/Aircraft/DISCOVER-AQ/Aerosol-TraceGas

TCEQ AutoGC data: https://www.tceq.texas.gov/cgi-bin/compliance/monops/agc_daily_summary.pl

NCEP Global Surface Observational Weather Data (DS461): http://rda.ucar.edu/datasets/ds461.0/

OMI HCHO column data: http://h2co.aeronomie.be

## Acknowledgements

We thank Heather Stewart and Mark Estes (TCEQ) for providing the ground special measurements at Conroe. The DIAL-HSRL mixed layer height data during the SEAC[4]RS were produced by Richard Ferrare, Johnathan Hair, and Amy Jo Scarino (NASA LaRC). The aircraft CO measurements were made by Glenn Diskin (NASA LaRC), and the DC-8 solar

radiation measurements during the SEAC[4]RS were made by Anthony Bucholtz (NRL). Isoprene measurements on board the NASA aircraft during SEAC[4]RS and DISCOVER-AQ were supported by the Austrian Federal Ministry for Transport, Innovation and Technology (bmvit) through the Austrian Space Applications Programme (ASAP) of the Austrian Research Promotion Agency (FFG). Markus Müller and Tomas Mikoviny are acknowledged for data acquisition and analysis. MH is grateful for the financial support from a NASA grant (NNX16AN39G) and NOAA GOES-R, as well as the technical

support from Li Fang, Jifu Yin (U Maryland), and the LIS/NUWRF teams at NASA GSFC.



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





## Figures

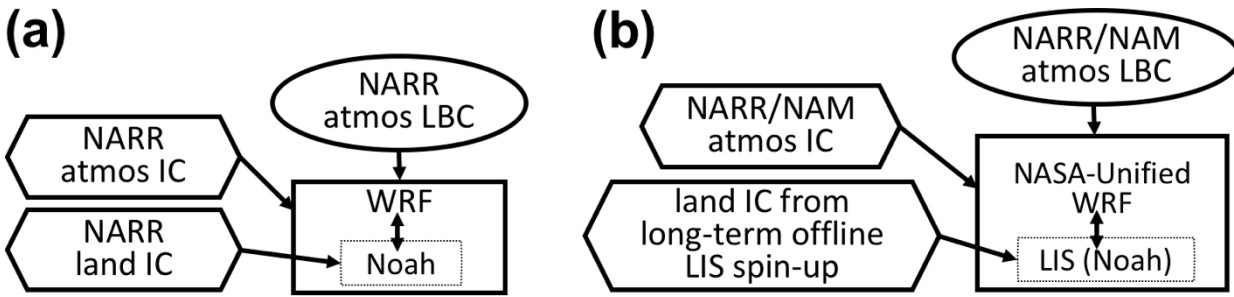

**Figure 1:** Illustration of the two (NU)WRF initialization methods compared in this study: (a) A usual method in which the land and atmospheric initial conditions (IC) and lateral boundary conditions (LBC) are downscaled from a coarse model output North American Regional Reanalysis (NARR); (b) The ctrl runs in which the land IC are from long-term offline LIS spin-up at the same grid resolutions as NUWRF, forced by highly resolved atmospheric forcing and precipitation data; the atmospheric IC and LBC from both NARR and the North American Mesoscale Forecast System (NAM) are used.



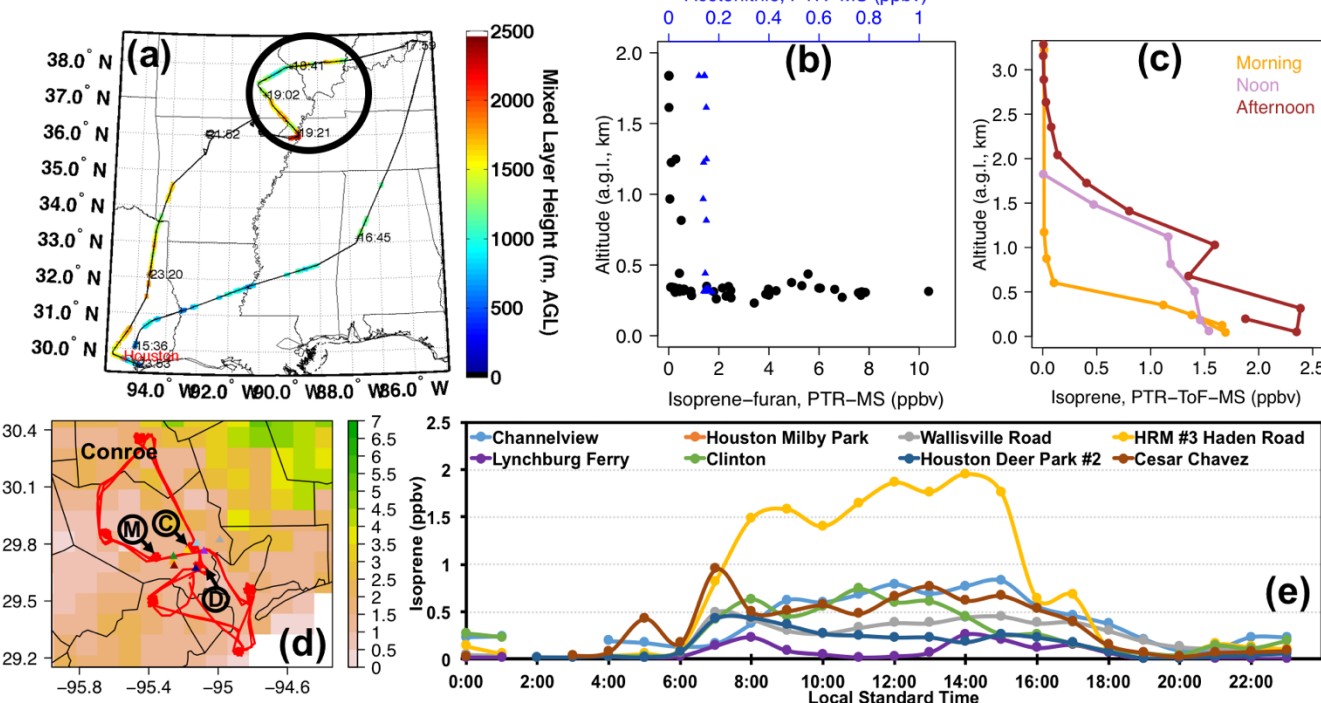

**Figure 2:** Measurements during the SEAC[4]RS and DISCOVER-AQ Houston campaigns on 11 September, 2013: (a) the DC-8 flight path colored by mixed layer height from the DIAL-HSRL instrument. Areas within the black circle indicate the "isoprene volcano" regions sampled at ~19 UTC (~13 local standard time). (b) Vertical profiles of the PTR-MS measured isoprene (black dots) and acetonitrile (blue triangles) at around the "isoprene volcano" regions. (c) Vertical profiles of the PTR-ToF-MS measured isoprene at the Conroe spirals in the Greater Houston area at different times of the day. (d) MODIS leaf area index (8-day mean with missing values filled with the monthly-mean) over the Greater Houston area in the 12 km NUWRF grid. The red solid line indicates the DISCOVER-AQ P-3B flight path. Colored triangles in Houston urban and ship channel areas denote the locations of surface sites that had hourly AutoGC isoprene measurements shown in (e). Note that only one AutoGC data point is available at the Milby Park site at 00 local standard time. Locations of the Conroe and urban/ship channel spirals (M: Moody Tower; C: Channelview; D: Deer Park, cited in Figure A1) are marked in black in (d).





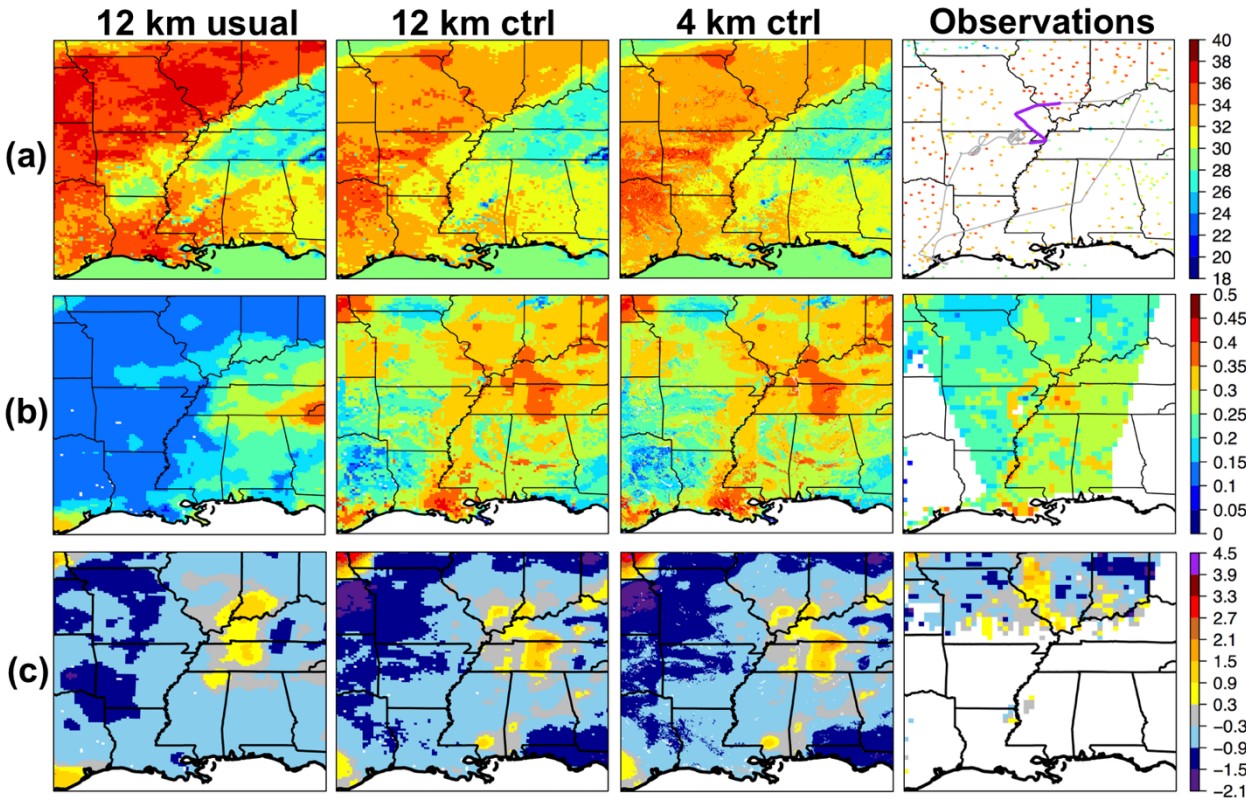

**Figure 3:** Evaluation of NUWRF's (a) surface air temperature in °C at ~13 local standard time and (b) surface soil moisture in $m^3/m^3$ and (c) normalized surface soil moisture at ~00 local standard time (the NUWRF initial time) over the central/southeastern US on 11 September, 2013. The NUWRF simulations are shown in the first three columns; Surface in-situ temperature observations (in the 12 km NUWRF grid) and the ESA CCI combined daily soil moisture product (0.25°×0.25°, only showing data with quality flag = 0) are shown in the right column. The normalized anomaly is defined in Eq. (3) in the text, and the right panel in (c) only shows the results at where data are available on >=80% of the days in September 2013.





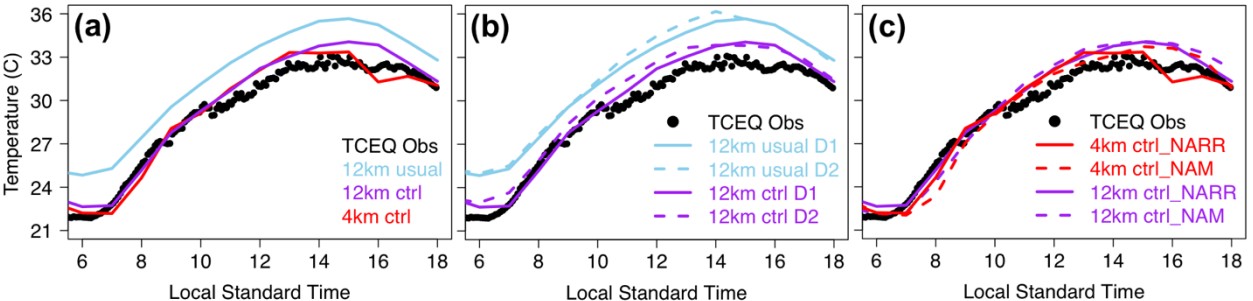

**Figure 4:** Comparing NUWRF's daytime surface air temperature with the 5-minute TCEQ surface in-situ measurements (black dots) at Conroe, TX, on 11 September, 2013: (a) Evaluating the impact of NUWRF land initialization and grid resolution; (b) Evaluating the impact of NUWRF simulation length, including the 1-day (D1) and 2-day (D2) forecasts, in the 12 km grid; (c) Evaluating the impact of NUWRF's atmospheric initialization (using NARR and NAM) in 12 km and 4 km grids.

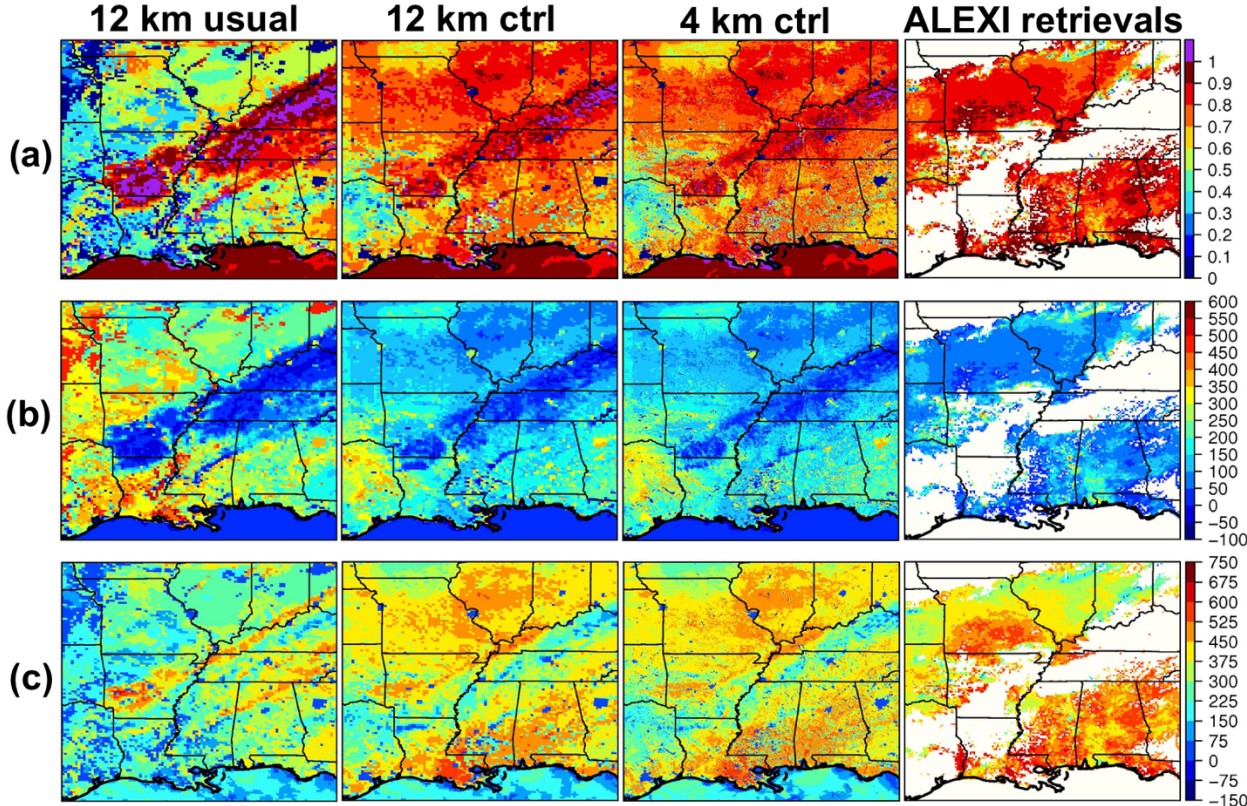

**Figure 5:** Comparing (a) evaporative ratio, unitless, defined as latent heat/(latent+sensible heat); (b) sensible heat in W/m$^2$, and (c) latent heat in W/m$^2$ from three NUWRF simulations (first three columns) with the NOAA operational 0.08° ALEXI retrievals (right), on 11 September, 2013.




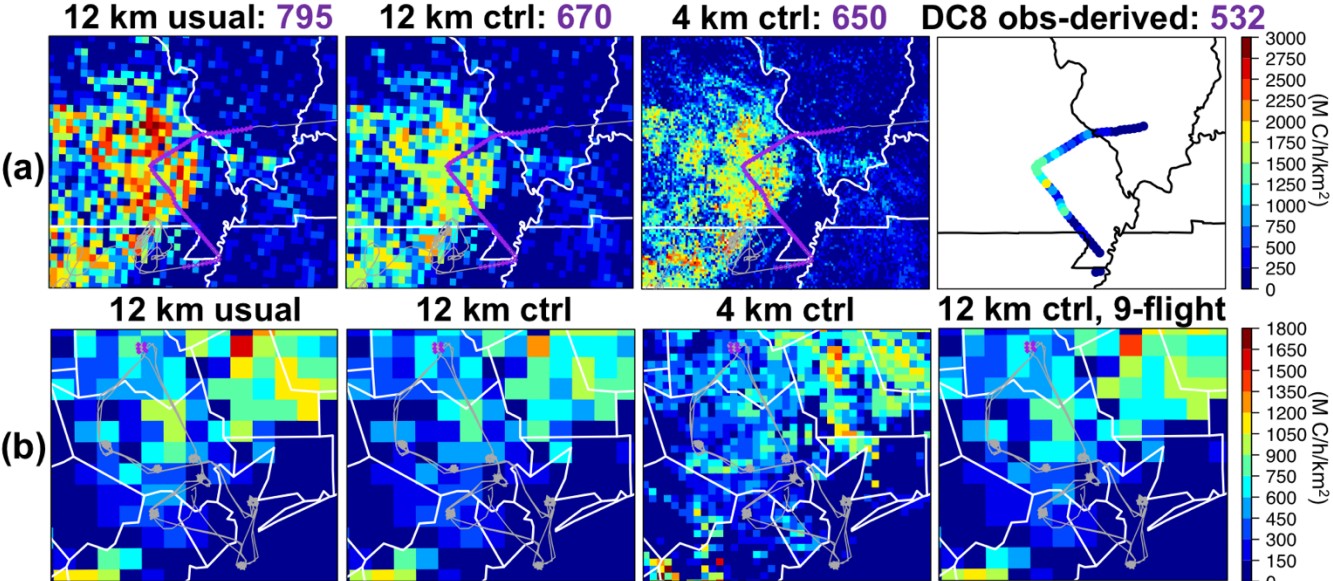

**Figure 6:** (a) Isoprene emissions around the "isoprene volcano" areas in Missouri based on NUWRF-MEGAN and aircraft observations at ~13 local standard time on 11 September, 2013. The mean values along the DC-8 flight path during 12:30-13:30 local standard time (in purple) are indicated in the figure captions. The open purple dots along the flight path refer to where isoprene data are missing or above the PBL. (b) Isoprene emissions in Houston, TX from NUWRF-MEGAN on 11 September, 2013 and on 9 DISCOVER-AQ flight days in September 2013, at local noontime. The DISCOVER-AQ P-3B flight path on 11 September, 2013 (note that the flight paths for all other flight days are similar to but not exactly the same as the 11 September's) is overlaid in grey and the Conroe samples at around the noontime are highlighted in purple.

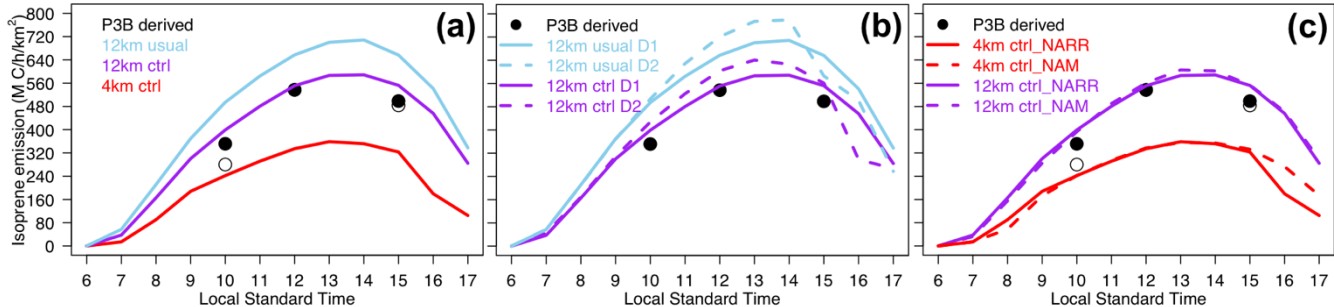

**Figure 7:** Comparing the NUWRF-MEGAN daytime isoprene emissions in Conroe on 11 September, 2013 with the observation-derived emissions. The black filled and open circles indicate the calculations based on PBLHs from the 12 km and 4 km NARR IC/LBC ctrl runs, respectively: (a) Evaluating the impact of NUWRF land initialization and grid resolution; (b) Evaluating the impact of NUWRF simulation length, including the 1-day and 2-day forecasts, in the 12 km grid; (c) Evaluating the impact of NUWRF's atmospheric initialization (using NARR and NAM) in 12 km and 4 km grids.





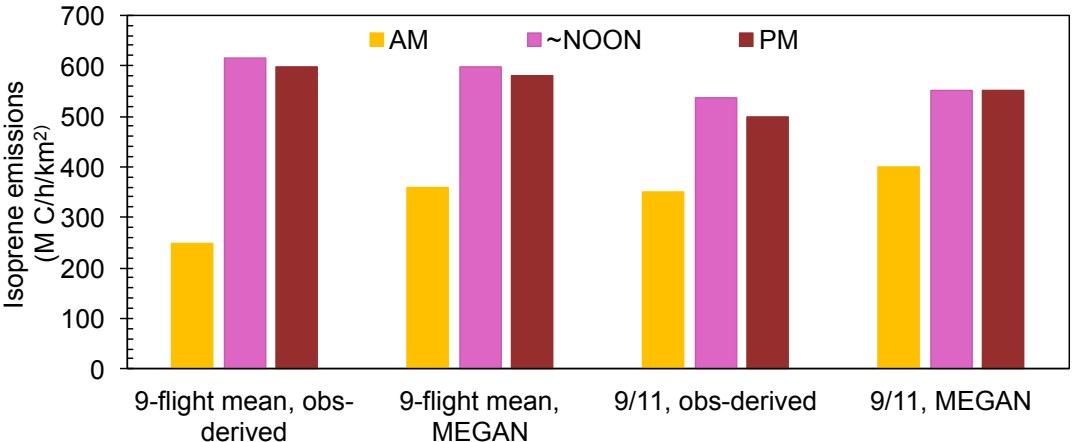

**Figure 8:** Temporal variability (AM: 15-16 UTC; ~noon: 18-19 UTC; PM: 20-21 UTC) of isoprene emissions in Conroe averaged on multiple DISCOVER-AQ flight days in September 2013 from NUWRF-MEGAN and aircraft observations, comparing with the 11 September, 2013 conditions. The multi-flight day mean P-3B isoprene, NAQFC CMAQ OH and NUWRF PBLH were used to derive the multi-day mean observation-based emissions.

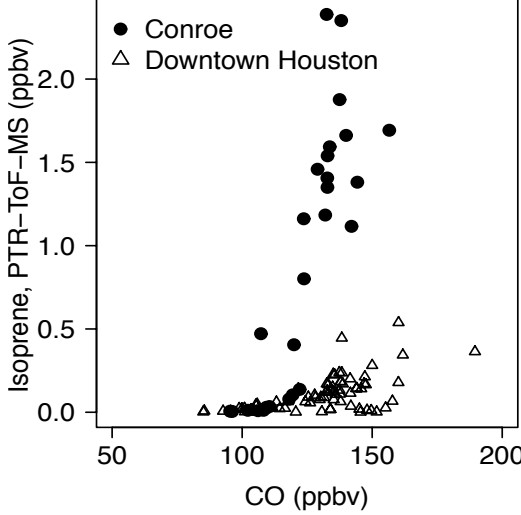

**Figure A1:** Scatterplot of the P-3B measured isoprene-carbon monoxide (CO) at the Conroe spirals (filled circles) and at three Downtown Houston/Ship Channel sites of Moody Tower, Deer Park and Channelview (open triangles), on 11 September, 2013. Locations of Moody Tower, Deer Park and Channelview are defined in Figure 2d. The CO measurements were taken using a Diode laser spectrometer measurements of CO, $CH_4$, $N_2O$ (DACOM) instrument with uncertainty of 2% or 2 ppbv.



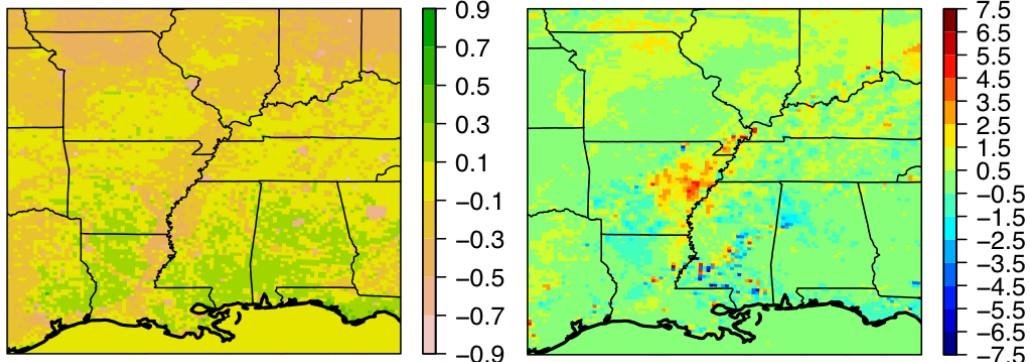

**Figure A2:** (left) The difference between daily near real-time GVF and the climatological monthly-mean GVF on 11 September, shown on the 12 km grid, and (right) the resulting differences in NUWRF usual runs' (12 km usual_veg-12 km usual, as defined in Table 1) surface air temperature in °C at 13 local standard time on this day.

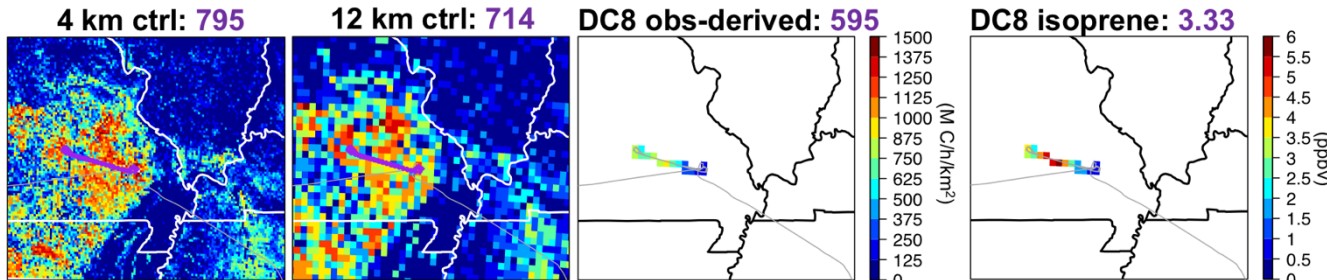

**Figure A3:** (Left three columns) Isoprene emissions around the "isoprene volcano" areas in Missouri from NUWRF-MEGAN at 14 local standard time and aircraft observations on 06 September, 2013. The mean values along the DC-8 flight path during 13:20-14:30 local standard time (when aircraft observations were made along four West-East transects) are indicated in purple in the figure captions. (Right column) Observed isoprene concentrations in ppbv along the DC-8 flight path during 13:20-14:30 local standard time, with the mean values shown in the figure captions. The observations and observation-derived isoprene emissions were averaged to the 12 km NUWRF grid for the plots. The 12 km ctrl NUWRF-MEGAN isoprene emissions overall have a slightly lower positive mean bias (~20%) from the observation-derived emissions than the 11 September result (~26%), whereas the positive biases of 4 km ctrl NUWRF-MEGAN emissions (~34%) are larger than the 11 September result (~22%). The largest overprediction occurs near the east side of the transects, where the biases are higher in the 4 km case than in the 12 km case. These biases are also shown by Wolfe et al. (2015) in their MEGAN emissions computed using a different meteorological input.





**Tables**

**Table 1:** Summary of all NUWRF simulations in this study

| Case Name | Horizontal resolution | Land initialization | Atmospheric initialization/ Lateral boundary conditions |
|---|---|---|---|
| 12 km usual(_NARR)[a] | 12 km | NARR | NARR |
| 12 km usual_veg(_NARR)[a,b] | 12 km | NARR | NARR |
| 12 km ctrl(_NARR)[a] | 12 km | LIS | NARR |
| 4 km ctrl(_NARR)[a] | 4 km | LIS | NARR |
| 12 km ctrl_NAM | 12 km | LIS | NAM |
| 4 km ctrl_NAM | 4 km | LIS | NAM |

[a]The *_NARR simulations are the focus of this study and the "(*_NARR)" part in the case names was often omitted in figures and texts. [b]Only shown in Figure A2 and briefly discussed in Section 3.1.1.

**Table 2:** NUWRF air temperature and PBLH performance along the DC-8 flight path (at where the isoprene data are available) around the "isoprene volcano" region in Missouri on 11 September, 2013. A 1.6 km PBLH, which was close to the 4 km/12 km simulated mean values (in italic/bold), was used to derive the emissions.

| NUWRF Case Name | PBLH (km), mean ± standard deviation | Air temperature RMSE (°C) |
|---|---|---|
| 4 km ctrl | *1.551*±0.304 | 0.735 |
| 12 km ctrl | *1.569±0.369* | 0.771 |
| 12 km usual | 2.190±0.630 | 2.241 |

**Table 3:** NUWRF-simulated median PBLH (km) and daytime (6-17 local standard time) surface air temperature performance at different times of 11 September, 2013, at around Conroe, TX. The bold italic numbers were used to derive emissions.

| NUWRF Case Name | PBLH (km) | | | Surface air temperature (°C)[a] | |
|---|---|---|---|---|---|
| | Morning | ~Noon | Afternoon | Mean bias | RMSE |
| 4 km ctrl | *0.666* | *1.467* | *1.752* | 0.159 | 0.747 |
| 12 km ctrl | *0.839* | *1.465* | *2.038* | 0.640 | 0.864 |
| 12 km usual | 1.195 | 1.746 | 2.337 | 2.473 | 2.507 |

[a]The two data points nearest to the full clocks from TCEQ 5-minute special ground observations

15  were averaged and compared with the hourly-recorded NUWRF output.