# Peer review of "Biogenic isoprene emissions driven by regional weather predictions using different initialization methods: Case studies during the SEAC4RS and DISCOVER-AQ airborne campaigns"

_Geoscientific Model Development, 2017_

## Referee Comment (RC1) · Anonymous Referee #1 · 21 Apr 2017

GENERAL COMMENTS

===============

The main goal of the manuscript by Huang et al. is to quantify the effect of different initial/boundary conditions on meteorological variables that drive isoprene emissions. The authors show convincingly how choice of land initial conditions (and in particular the time used for initialisation) have a much larger impact on atmospheric temperatures, and therefore isoprene emissions, the atmospheric conditions.

This is important work that should be of interest to both the meteorological and atmospheric chemistry modelling communities. The work is well within the scope of GMD. My main concerns are around presentation – I find the text very hard to read, the title uninformative. As a result, the important and interesting messages get lost amongst the details. Substantial editing is required before publication in GMD. More details below.

Title – The title is long and detailed, but to me it doesn't accurately represent what is to come in the text. As someone who doesn't use the NASA-Unified WRF model, I wouldn't read this paper, but having now read it I realise it is relevant to my work after all! I encourage the authors to revise it to inform the reader that the focus is on the impacts of different choices of land and atmosphere initial conditions on ability to simulate biogenic emissions. I don't think the choice of SEAC4RS and DISCOVER-AQ is relevant enough to include in the title – these just happened to be the most relevant evaluation observations available.

Readability – The text throughout is dense, often hard to follow, and full of grammatical and spelling errors. I highlight some but not all of these below. I suggest a very careful read-through and edit (if not by the authors, then perhaps by a professional editor). There are a very large number of acronyms used throughout, and since this work should appeal to multiple modelling communities, it may be useful where possible to use a description rather than an acronym. An appendix listing all acronyms (and what they refer to, since in some cases model names are not intuitive) would also be helpful/

Abstract – There are too many acronyms in the abstract, making it hard to follow. Most of these are not needed until the main text. For example, lines 19-20, at this stage the reader doesn't need to know the specific land surface model or reanalysis data set used. Just stating ". . . demonstrating that initialising the input land surface model using a coarser resolution dataset led to significant positive biases. . ." would be much clearer. Same applies elsewhere. Also the sentence "This study emphasizes. . . chemical data assimilation" (lines 29-32) is very long and could be split into two for clarity.

Introduction – Much of the introduction doesn't seem well suited to the work presented in this particular manuscript. For example, the entire first paragraph that discusses isoprene impacts on ozone seems irrelevant as there is no further mention (or simulation) of ozone. As this is a GMD paper, the focus on O3 seems irrelevant (and in any case, the references seem spotty and cherry-picked to only discuss those that have shown large responses to isoprene – some others off the top of my head include Wu et al. 2008 (doi: 10.1029/2007JD008917), Millet et al., 2016 (doi:10.1021/acs.est.5b06367)). In the second paragraph, there is too much detail about MEGAN that can wait until the model description (for example, "on flexible scales", "MEGAN computes emissions based on. . .". I suggest reorganising to start with the current paragraph 3, then following with some of paragraph 2 (with emphasis on uncertainties/errors in MEGAN), then paragraph 4.

"Usual" and "Control" – the choice of language to describe the simulations is very confusing. What is described here as "usual" is what most authors mean when they say "control" (i.e. the control simulation is the normal or base method, without modifications. So using the word "control" to describe the simulations where something new/different is done is very counterintuitive. It took me until the 2nd reading of the paper to actually work out which simulations used something new. Suggest "usual" becomes "control" and "control" becomes "sensitivity".

3.3 Uncertainty discussions – while I appreciate the attempt to discuss remaining uncertainties, I find the current version in Section 3.3 doesn't add much value. It needs to include some discussion (i.e. literature based) of what the expected impacts of each of these things would be (I'm mainly referring to the NUWRF-MEGAN section here). I also find (d) in this section completely meaningless ("other limitations") – if they aren't going to be stated or explained, then why bother mentioning them?

Fig. 1 – I like the idea of an overview figure, but find this one hard to understand. Suggest adding resolution to the figure, and separating (b) to show the different types of sensitivity simulations that were done (perhaps including names from the table).

**SPECIFIC COMMENTS**

===============

Pg 2, line 21-22: Also Zeng et al., 2015 (doi:10.5194/acp-15-7217-2015); Emmerson et al., 2016 (doi:10.5194/acp-16-6997-2016)

Pg 2, line 28-29: "Much less has been done..." – see Zheng et al., 2015 (doi:10.5194/acp-15-8559-2015), Bauwens et al., 2016 (doi:10.5194/acp-16-10133-2016)

Pg 3, line 18: "key variables" – state them here

Pg 5, line 29 – Pg 6, line 18: I got very lost in this paragraph – which is an important one for understanding what was actually done! I think it needs to be rewritten to first highlight what the different types of simulations are (in simplified form), followed by one paragraph to describe the basic simulation and another (or two) to describe the modifications in the other simulations.

Pg 7, first paragraph: reference Toon et al. (doi: 10.1002/2015JD024297)

Pg 7, line 25 and elsewhere: I think Figs. A1-A3 belong in a Supplement rather than an Appendix as there is no associated text and they are not related to one another.

Pg 8, line 14: "Close to the estimated OH concentrations... 2006" – what are these? Please provide the value here to allow the reader to make the comparison

Pg 10, line 22: DIAL-HSRL derived PBLH is mentioned here but not show or discussed, besides saying the model is "closer to the reality" of it. More discussion or plots needed to justify that statement.

Pg 11, line 32: 30% seems to me like a big bias, seems like this requires some discussion

Pg 13-14, Section 3.2.2: I don't get anything out of this section and am unclear what I

am meant to take away. I suggest moving it to a Supplement and referencing briefly in the main text.

Pg 14, line 27: "random" – is it really a random error? It seems like if there were even reasonably consistent wind directions in a given location, this would be a systematic error. . .

Pg 15, line 9: "result in Wolfe et al. (2015) of 8+/1 mg/m2/h)". This comparison value is given in a completely different unit from the one shown in the figure (and it is also hard to compare when the figure and text are not co-located). Please provide the value in the text and make sure the units of the calculated value and the Wolfe et al. value are the same (doesn't matter which).

Pg 15, line 16: "LIS simulation" – explain what type of model LIS is for the reader that skips straight to the conclusion

Pg 16, line 8: "self spin-up method" – what is this? Needs some explanation

Fig 3: Not obvious why (b) and (c) show different coverage in the observations, and the choice of >=80% is not justified. Either justify or (preferable) just show all data in (c)

Fig 8: Would be easier to understand/interpret is bars were grouped by AM, noon, and PM (then colored by different dates/campaigns)

TECHNICAL COMMENTS

==============

As noted previously, there are many errors of grammar, spelling, or poor word choice. I point out some but not all of them here.

Pg 1 line 25: "modify" -> "reduce"

Pg 1, line 29: "resulted" -> "resulting"

Pg 2, line 5: "50% of reduction" -> "50% reduction"
Pg 2, line 14: "over the NA" -> "over NA"

Pg 3, line 17: "experimented" not right word, perhaps "tested"

Pg 4, line 6: should "LAI" actually be "gamma_LAI" (with symbol)?

Pg 4, lines 7-8: "which needs to be better understood" – irrelevant to this work, delete

Pg 4, line 16: reference the actual data rather than a blog post

Pg 4, line 27-28: this sentence references Fig 2 before Fig 1 – either need to rearrange the figure or (preferable) remove this sentence and put it in the section where Fig 2 is described

Pg 5, line 13: "Same as in. . ." -> "As in. . ."

Pg 5, line 14: "four-soil layer" - should this be "four-layer soil"?

Pg 5, line 15: "widely used" – any references?

Pg 5, line 26: "full clocks" – what does this mean? I've never seen this term.

Pg 6, line 2: "Same as in. . ." -> "As in. . ."

Pg 6, line 5: "representing" -> "represent:

Pg 6, line 7: "focused" -> "focus"

Pg 6, line 26: "rate coefficients with OH" – does this mean "rate coefficient of isoprene with OH"?

Pg 8, lines 27, 28: "were" "was" (twice)

Pg 9, line 6-7: "Long-term soil moisture changes. . . p72-73)" – this sentence is irrelevant to the work presented here, delete.

Pg 9, line 12: "at where" -> "where"

Pg 10, lines 10-13: "The magnitudes. . . daily minima" – long and confusing sentence,

suggest splitting into two

Pg 10, lines 20-22: "RMSEs . . . uncertain) – long and confusing sentence, suggest splitting into two

Pg 11, line 10: Fig 5 referenced before Fig 4 – reorder these figures

Pg 12, line 3: "activate" -> "active"

Pg 12, line 26: "weakest" -> "weaker"

Pg 15, line 21: "resulted" -> "resulting"

Pg 16, line 10: "Experimenting simulations" -> "Experiments using simulations"

Fig 2a: colors are hard to see; how about plotting the color on top of the black line rather than below?

Fig 2d: colorbar missing label

---

## Referee Comment (RC2) · Anonymous Referee #2 · 31 May 2017

**Linkages between land initialization of the NASA-Unified WRF v7 and biogenic isoprene emission estimates during the SEAC4RS and DISCOVER-AQ airborne campaigns**
**Huang et al., GMDD, 2017**

**General Description of manuscript and recommendation:**

The authors use aircraft, surface, and satellite observations to estimate isoprene emissions at high spatial resolution over the southeastern US. This study demonstrates that long-term initialization of land cover substantially improves modelled meteorology relevant for estimating isoprene emissions and boundary layer dynamics. The work is relevant to GMD, but in reviewing the manuscript I had to spend a considerable amount of time sifting through errors and poor grammar. This hampered my ability to evaluate the quality of the science, the interpretation of the results, and overall conclusion of the study. I suggest major revisions be applied to the manuscript so that it is more accessible to the GMD readership.

**General Comments:**

The manuscript is not carefully edited. There are many errors (e.g. page 6, line 27 exponential "12" should be "-12"), many misuses of punctuation (e.g. semicolons and colons in the sentence on page 6, lines 4-10), and often lengthy hard-to-follow sentences (e.g. page 2, lines 7-10; page 2, lines 11-14; page 6, lines 4-10). All of these issues stand in the way of communicating the science and leave the reader confused. Some examples of these are provided, but please read through the manuscript carefully to identify and address these issues.

There are many instances where "the" is used when it is not necessary, e.g. the is not needed in "the sunset" or "the sunrise" (page 7, lines 27-28). "The" is only needed when the noun is specific or particular, for example there is only one NUWRF-MEGAN model (page 8, line 2), one North America (page 2, line 14) and one sunrise (page 7, line 27), so "the" is not needed.

Figures are not presented in order in the text. The authors first introduce Figure 2(d) then goes on to mention Figure 1. Reorder figures to reflect the order in which they appear in the text.

There is unnecessary repetition, in particular in Section 3.1. That SEAC4RS has no/minimal biomass burning interference (page 10, lines 2-3) is already stated in Section 2.2.1, as is the limited contribution of anthropogenic VOC interference to measurements in Conroe (page 10, lines 7-10). No need to state all this again in Section 3.1.

Possessive is not necessary when describing data from a model or measurement platform. For example, replace "NUWRF's day time surface air temperature" with "NUWRF day time surface air temperature". There are many other instances where apostrophes are used, but aren't required. Please identify and correct these.

The equation used to infer isoprene emissions relies on OH concentration and boundary layer height as input. I would like to see some discussion and evaluation of the diurnal variability of these parameters, as these are used as input in Eq. (2) to estimate isoprene emissions from the isoprene concentration observations.

**Specific Comments:**
Page 4, line 4: What emissions come from the atmosphere to the canopy? Do the authors mean the emissions consumed/deposited within the canopy (term $\rho$ in Eq. (1))? If this is not considered in the emission model used in this study, then please clarify that this value is set to 1 to avoid confusion.

Page 5, lines 15-18: What is the difference in land cover between the IGBP-derived land cover that is used in this study and the default used in MEGAN to justify using an updated land cover map? Some discussion of how using this updated land cover impacts isoprene emission modelling is needed.

Page 5, lines 23-24: What is "the urban surface option"?

Page 5, line 26; Table 3 footnote: What is "full clocks"? This isn't standard terminology. Rather describe this in a way that can be understood.

Page 7, lines 4-5: Why mention the August western US flights? Seems it has no relevance to this study and so can be removed.

Page 11, line 32: Is "vastly similar" correct? In the next clause, the authors state that the difference is >30%.

Page 12, lines 29-31: Why does resolution induce a difference in isoprene emissions in the atmospheric initialization sensitivity test in Figure 7c) and not in Figure 4c)?

Page 16, lines 1-2: Please provide references to back up the statement that many model comparison studies don't adequately assess the impact of model inputs.

Figure 3: Please increase the size of the points in the Figure 3a) Observations panel so that the reader can easily compare the observations and model or instead show a scatterplot of the model versus the observations and include regression statistics.

Figure 3 caption: Please say where the temperature observations are from. Are these NCEP?

---

## Author Comment (AC1) · 23 Jun 2017

We thank the careful review by both reviewers. Both of them mentioned that the paper was relevant to GMD, but needed to be edited more carefully. They offered suggestions on improving its readability. We have substantially edited the paper with all their comments addressed. Please see below our point-by-point response (in blue) to their general and specific comments (in black).

**Response to Referee 1's comments**

GENERAL COMMENTS
===============
The main goal of the manuscript by Huang et al. is to quantify the effect of different initial/boundary conditions on meteorological variables that drive isoprene emissions. The authors show convincingly how choice of land initial conditions (and in particular the time used for initialisation) have a much larger impact on atmospheric temperatures, and therefore isoprene emissions, the atmospheric conditions.

This is important work that should be of interest to both the meteorological and atmospheric chemistry modelling communities. The work is well within the scope of GMD. My main concerns are around presentation – I find the text very hard to read, the title uninformative. As a result, the important and interesting messages get lost amongst the details. Substantial editing is required before publication in GMD. More details below.

Title – The title is long and detailed, but to me it doesn't accurately represent what is to come in the text. As someone who doesn't use the NASA-Unified WRF model, I wouldn't read this paper, but having now read it I realise it is relevant to my work after all! I encourage the authors to revise it to inform the reader that the focus is on the impacts of different choices of land and atmosphere initial conditions on ability to simulate biogenic emissions. I don't think the choice of SEAC4RS and DISCOVER-AQ is relevant enough to include in the title – these just happened to be the most relevant evaluation observations available.

The revised title is: "Biogenic isoprene emissions driven by regional weather predictions using different initialization methods: Case studies during the SEAC$^4$RS and DISCOVER-AQ airborne campaigns".

We noticed that GMD has a requirement for "model description papers" of *"The main paper must give the model name and version number (or other unique identifier) in the title"* (http://www.geoscientific-model-development.net/about/manuscript_types.html), and to be safe we included "NASA-Unified WRF v7" in the previous title although this is a "model evaluation paper". We have removed the model name and version because we do agree with the reviewer that the methodology in this paper may interest other regional modelers. The model name and version can still be found in the abstract.

The revised primary title has been shortened with the campaign names moved to the secondary title. The campaign names in the title serve as identifiers of the study region/period as well as the datasets used. This usage is often seen in publications, and has been recognized to be informative.

Readability – The text throughout is dense, often hard to follow, and full of grammatical and spelling errors. I highlight some but not all of these below. I suggest a very careful read-through and edit (if not by the authors, then perhaps by a professional editor). There are a very large number of acronyms used throughout, and since this work should appeal to multiple modelling communities, it may be useful where possible to use a description rather than an acronym. An appendix listing all acronyms (and what they refer to, since in some cases model names are not intuitive) would also be helpful/

The readers would take extra efforts to digest this paper, in part due to its cross-disciplinary nature, which actually makes this study unique. The paper has been substantially edited, with all of the language-related specific and technical comments addressed. We took your suggestion to add a categorized list of acronyms in the appendix-For each model related item, a short description is included; For each instrument/product related item, the measured variables we used for this study are included. We also deleted unnecessary definitions of acronyms (i.e., those only used once throughout the paper).

Abstract – There are too many acronyms in the abstract, making it hard to follow. Most of these are not needed until the main text. For example, lines 19-20, at this stage the reader doesn't need to know the specific land surface model or reanalysis data set used. Just stating ". . . demonstrating that initialising the input land surface model using a coarser resolution dataset led to significant positive biases. . ." would be much clearer. Same applies elsewhere. Also the sentence "This study emphasizes. . . chemical data assimilation" (lines 29-32) is very long and could be split into two for clarity.

These are good suggestions. We are now not specific about the land surface model (Noah) and datasets (NARR or NAM) used in the abstract. The sentence near L29-32 has been split and now reads as: "This study emphasizes the importance of proper land initialization to the coupled atmospheric weather modeling and the follow-on emission modeling. We anticipate it to be also critical to accurately representing other processes included in air quality modeling and chemical data assimilation."

Introduction – Much of the introduction doesn't seem well suited to the work presented in this particular manuscript. For example, the entire first paragraph that discusses isoprene impacts on ozone seems irrelevant as there is no further mention (or simulation) of ozone. As this is a GMD paper, the focus on O3 seems irrelevant (and in any case, the references seem spotty and cherry-picked to only discuss those that have shown large responses to isoprene – some others off the top of my head include Wu et al. 2008 (doi: 10.1029/2007JD008917), Millet et al., 2016 (doi:10.1021/acs.est.5b06367)). In the second paragraph, there is too much detail about MEGAN that can wait until the model description (for example, "on flexible scales", "MEGAN computes emissions based on. . .". I suggest reorganising to start with the current paragraph 3, then following with some of paragraph 2 (with emphasis on uncertainties/errors in MEGAN), then paragraph 4.

This study aims to improve the biogenic emission estimates as that may further improve ozone (a regulated pollutant) air quality modeling. There are indeed many publications on isoprene-ozone relationships, but we chose to cite in the opening paragraph several modeling studies over the similar focus regions to this study's. Several long sentences in this paragraph have been reworded. As evaluating isoprene emissions is a primary goal in this work, we introduce MEGAN first including its sensitivity to weather inputs, followed by the contents on weather modeling.

In the conclusion, the findings in this study are connected with air quality modeling: "We anticipate that improved weather fields using the better land initialization approach will also benefit the representation of the other processes (other weather-dependent emission calculations, transport, transformation, deposition) included in air quality modeling, and therefore can help reduce uncertainty in the simulated chemical fields."

"Usual" and "Control" – the choice of language to describe the simulations is very confusing. What is described here as "usual" is what most authors mean when they say "control" (i.e. the control simulation is the normal or base method, without modifications. So using the word "control" to describe the simulations where something new/different is done is very counterintuitive. It took me until the 2nd reading of the paper to actually work out which simulations used something new. Suggest "usual" becomes "control" and "control" becomes "sensitivity".

We added and modified some sentences in Section 2.1.3 to clarify the naming criteria. "Usual" means the method used "in default and many WRF applications". "Control" was chosen to be consistent with the usage in hydrological modeling community. It means that high quality forcing data were used in the offline LIS simulation. Such simulation is often compared with "open-loop" and "assimilated" LIS runs, in which low-quality precipitation data were used, to assess the usefulness of soil moisture data assimilation.

An example: Hain, C. R., W. T. Crow, M. C. Anderson, and J. R. Mecikalski (2012), An ensemble Kalman filter dual assimilation of thermal infrared and microwave satellite observations of soil moisture into the Noah land surface model, Water Resour. Res., 48, W11517, doi:10.1029/2011WR011268.

The definitions and descriptions of these various cases can be found at multiple places of this paper (Section 2.1.3, Table 1, Figure 1), so the readers won't miss them.

3.3 Uncertainty discussions – while I appreciate the attempt to discuss remaining uncertainties, I find the current version in Section 3.3 doesn't add much value. It needs to include some discussion (i.e. literature based) of what the expected impacts of each of these things would be (I'm mainly referring to the NUWRF-MEGAN section here). I also find (d) in this section completely meaningless ("other limitations") – if they aren't going to be stated or explained, then why bother mentioning them?

The list of error sources of NUWRF-MEGAN serves as a summary for the readers' quick references. Point a) has been expanded, and points b)-c) were quantitatively discussed in previous sections. It'd be challenging to quantify impact from points a) and d) for this specific case, but we wanted to explicitly point them out, and they can be good topics of future studies. Although we are not specific about "other limitations", point d) is included to help strengthen the point that having more confidence in the weather inputs would help better determine and quantify those impacts.

Fig. 1 – I like the idea of an overview figure, but find this one hard to understand. Suggest adding resolution to the figure, and separating (b) to show the different types of sensitivity simulations that were done (perhaps including names from the table).

This figure is designed to illustrate the different land and atmospheric initialization approaches we compared in this study. The squares, circles, and hexagons represent (NU)WRF model, its

atmospheric lateral boundary conditions, and initial conditions, respectively. We now include resolutions in the figure captions, and also have added a column in Table 1 to connect this figure with the different cases listed in the table.

SPECIFIC COMMENTS
===============
Pg 2, line 21-22: Also Zeng et al., 2015 (doi:10.5194/acp-15-7217-2015); Emmerson et al., 2016 (doi:10.5194/acp-16-6997-2016)
Thanks for the suggestions. We added Emmerson et al. (2016) which fits better into this context.

Pg 2, line 28-29: "Much less has been done. . ." – see Zheng et al., 2015 (doi:10.5194/acp-15-8559-2015), Bauwens et al., 2016 (doi:10.5194/acp-16-10133- 2016)
These are very good global-scale studies. We added "at multiple spatial-temporal scales" in this sentence. Smaller-scale analyses like this study are novel and would be more relevant to air quality modeling and management. The "less" word is relevant to the significant attention given to temperature and radiation. Both the direct ($\gamma_{SM}$) and indirect impacts (atmosphere-land coupling) are mentioned in the following sentences.

Pg 3, line 18: "key variables" – state them here
Added.

Pg 5, line 29 – Pg 6, line 18: I got very lost in this paragraph – which is an important one for understanding what was actually done! I think it needs to be rewritten to first highlight what the different types of simulations are (in simplified form), followed by one paragraph to describe the basic simulation and another (or two) to describe the modifications in the other simulations.
We broke this long paragraph into separate ones (points a-c). Long sentences were shortened. Figure 1 was also revised according to your suggestion, and its a-c panels now match the three points here.

Pg 7, first paragraph: reference Toon et al. (doi: 10.1002/2015JD024297)
Added.

Pg 7, line 25 and elsewhere: I think Figs. A1-A3 belong in a Supplement rather than an Appendix as there is no associated text and they are not related to one another.
We keep Figure A1, which is now included in Figure 2. Figures A2 and A3 were moved to the SI.

Pg 8, line 14: "Close to the estimated OH concentrations. . . 2006" – what are these? Please provide the value here to allow the reader to make the comparison
Reading from Figure 7 of Warneke et al. (2010), at the beginning and near the end of that flight (near Houston), the estimates fell within $2\text{-}6 \times 10^6$ molecule/cm$^3$, so we added "of approximately 2-$6 \times 10^6$ molecule/cm$^3$..."

Pg 10, line 22: DIAL-HSRL derived PBLH is mentioned here but not show or discussed, besides saying the model is "closer to the reality" of it. More discussion or plots needed to justify that statement.

We added "Figure 2a" to be more specific. At the beginning of Section 3.1, we introduced that "..where mixed layer heights indicated by the DIAL-HSRL instrument were mostly below 2 km", so the ctrl run results in Table 2, which are described here, appear to be closer to the DIAL-HSRL data.

Pg 11, line 32: 30% seems to me like a big bias, seems like this requires some discussion
We compared this bias with literature in the following sentence, and some discussions were placed in Section 3.3, NUWRF point b. More sensitivity analyses and discussions on this would be divergent from the focus of this paper.

Pg 13-14, Section 3.2.2: I don't get anything out of this section and am unclear what I am meant to take away. I suggest moving it to a Supplement and referencing briefly in the main text.
This section is already very concise. The paper mainly focuses on a short period during two field campaigns, and decadal satellite data help contrast this period to conditions in the other years so that the readers would get a better sense about the general conditions and temporal variability over this region. The satellite based analysis also indicates the relationships between soil moisture, vegetation and HCHO/isoprene. Related sentences in the last paragraph of Section 1 have been modified.

Pg 14, line 27: "random" – is it really a random error? It seems like if there were even reasonably consistent wind directions in a given location, this would be a systematic error. . .
We removed "random".

Pg 15, line 9: "result in Wolfe et al. (2015) of 8+/1 mg/m2/h)". This comparison value is given in a completely different unit from the one shown in the figure (and it is also hard to compare when the figure and text are not co-located). Please provide the value in the text and make sure the units of the calculated value and the Wolfe et al. value are the same (doesn't matter which).
We now use the converted values from the results in Wolfe et al. (2015).

Pg 15, line 16: "LIS simulation" – explain what type of model LIS is for the reader that skips straight to the conclusion
Added: "a flexible land surface modeling and data assimilation framework"

Pg 16, line 8: "self spin-up method" – what is this? Needs some explanation
This is a method suggested by Angevine et al. (2014). Before this citation, we added: "i.e., running the model for a certain spinup period (e.g., a month) at least once, cycling its own soil variables, to allow the land variables to develop appropriate spatial variability".

Fig 3: Not obvious why (b) and (c) show different coverage in the observations, and the choice of >=80% is not justified. Either justify or (preferable) just show all data in (c)
Unlike model data, the daily CCI data do not always cover everywhere in September 2013, so in (c) we meant to compare the results over the region with better satellite data coverage. To improve the presentation, we now show data in (c) without screening by the data size. However, to help the readers understand these results, we added Figure S2 to include the usable data size and uncertainty (defined as the standard deviation of data from multiple sensors) of the 11 September daily combined soil moisture data.

Fig 8: Would be easier to understand/interpret is bars were grouped by AM, noon, and PM (then colored by different dates/campaigns)
Revised.

TECHNICAL COMMENTS
===============
As noted previously, there are many errors of grammar, spelling, or poor word choice. I point out some but not all of them here.
Pg 1 line 25: "modify" -> "reduce"
Since we are not specific about which atmospheric initialization (NARR or NAM) worked better than the other, "modify" would be more appropriate than "reduce".

Pg 1, line 29: "resulted" -> "resulting"
Done.

Pg 2, line 5: "50% of reduction" -> "50% reduction"
Done.

Pg 2, line 14: "over the NA" -> "over NA"
Done.

Pg 3, line 17: "experimented" not right word, perhaps "tested"
Done.

Pg 4, line 6: should "LAI" actually be "gamma_LAI" (with symbol)?
The original form is consistent with the usage in Guenther et al. (2012) and Sindelarova et al. (2014).

Pg 4, lines 7-8: "which needs to be better understood" – irrelevant to this work, delete
Done.

Pg 4, line 16: reference the actual data rather than a blog post
We now report the weekly data along with the data source from the Scripps.

Pg 4, line 27-28: this sentence references Fig 2 before Fig 1 – either need to rearrange the figure or (preferable) remove this sentence and put it in the section where Fig 2 is described
Thanks for catching this. This information is now first introduced in Section 3.1. We also split the previous Figure 2 to improve its readability.

Pg 5, line 13: "Same as in. . ." -> "As in. . ."
Done.

Pg 5, line 14: "four-soil layer" - should this be "four-layer soil"?
Pg 5, line 15: "widely used" – any references?

This sentence has been rewritten, with two references added on scientific and NOAA operational use of the Noah LSM.

Pg 5, line 26: "full clocks" – what does this mean? I've never seen this term.
Rewritten as: "..recorded hourly at 00:00 (minute:second).."

Pg 6, line 2: "Same as in. . ." -> "As in. . ."
Done.

Pg 6, line 5: "representing" -> "represent:
Done.

Pg 6, line 7: "focused" -> "focus"
Done.

Pg 6, line 26: "rate coefficients with OH" – does this mean "rate coefficient of isoprene with OH"?
Yes, and we corrected 12 to -12.

Pg 8, lines 27, 28: "were" "was" (twice)
The "were" in both sentences were changed to "was".

Pg 9, line 6-7: "Long-term soil moisture changes. . . p72-73)" – this sentence is irrelevant to the work presented here, delete.
Removed.

Pg 9, line 12: "at where" -> "where"
Changed to "over the regions where"

Pg 10, lines 10-13: "The magnitudes. . . daily minima" – long and confusing sentence, suggest splitting into two
It now reads as: "The magnitudes of the downtown aircraft isoprene measurements were slightly lower than most of the nearby surface measurements during the daytime (Figure 2f). Measured surface isoprene levels during the daytime were ~twice as high as during the nighttime (~0.2-0.3 ppbv) when biogenic emissions are at their daily minima."

Pg 10, lines 20-22: "RMSEs . . . uncertain) – long and confusing sentence, suggest splitting into two
It now reads as: "Root Mean Square Errors (RMSEs) of the modeled air temperature from the ctrl runs are ~1.5 °C lower than the 12 km usual run-produced. The PBLHs from the ctrl run were thinner (~0.6 km on average) and less spatially variable. They may be closer to the reality referring to the DIAL-HSRL data in Figure 2a, which can also be uncertain."

Pg 11, line 10: Fig 5 referenced before Fig 4 – reorder these figures
Figure 4 was first referenced in Pg 10, before this line.

Pg 12, line 3: "activate" -> "active"
Done.

Pg 12, line 26: "weakest" -> "weaker"
Done.

Pg 15, line 21: "resulted" -> "resulting"
Done.

Pg 16, line 10: "Experimenting simulations" -> "Experiments using simulations"
Done.

Fig 2a: colors are hard to see; how about plotting the color on top of the black line rather than below?
Fig 2d: colorbar missing label
Figure 2 has been split to improve its readability. A label "LAI $(m^2/m^2)$" was added near the colorbar of the previous Figure 2d (current 2c).

**Response to Referee 2's comments**

**General Comments:**
The manuscript is not carefully edited. There are many errors (e.g. page 6, line 27 exponential "12" should be "-12"), many misuses of punctuation (e.g. semicolons and colons in the sentence on page 6, lines 4-10), and often lengthy hard-to-follow sentences (e.g. page 2, lines 7-10; page 2, lines 11-14; page 6, lines 4-10). All of these issues stand in the way of communicating the science and leave the reader confused. Some examples of these are provided, but please read through the manuscript carefully to identify and address these issues.

The readers would take extra efforts to digest this paper, in part due to its cross-disciplinary nature, which actually makes this study unique. The paper has been substantially edited, with all of the comments by both reviewers addressed. We believe the readability of the paper has been significantly improved.

The exponential 12 was corrected to "-12" and we apologize for any confusion due to this typo.

The sentences near lines 4, page 6 have been rewritten per Referees' suggestion. The semicolon was used to separate point a) and point b), but now points a)–c) are written as separate paragraphs. Figure 1 was also revised according to Referee 1's suggestion, and its a-c panels match points a-c here.

Lengthy sentences pointed out by both reviewers were reworded.

There are many instances where "the" is used when it is not necessary, e.g. the is not needed in "the sunset" or "the sunrise" (page 7, lines 27-28). "The" is only needed when the noun is specific or particular, for example there is only one NUWRF-MEGAN model (page 8, line 2), one North America (page 2, line 14) and one sunrise (page 7, line 27), so "the" is not needed.

We agree and removed those unnecessary "the"s.

Figures are not presented in order in the text. The authors first introduce Figure 2(d) then goes on to mention Figure 1. Reorder figures to reflect the order in which they appear in the text.

This issue was also brought up by Referee 1. The figures have been rearranged in order. Also, Figure 2 has been split to improve its readability. Figure A1 is now Figure 2e, and Figures A2-A3 are now in the SI.

There is unnecessary repetition, in particular in Section 3.1. That SEAC4RS has no/minimal biomass burning interference (page 10, lines 2-3) is already stated in Section 2.2.1, as is the limited contribution of anthropogenic VOC interference to measurements in Conroe (page 10, lines 7-10). No need to state all this again in Section 3.1.

The sentence in Section 2.2.1 of "Furan is found in significant concentrations only in biomass burning plumes and no enhancement in the biomass burning tracer acetonitrile was observed in our case studies (details in Section 3.1)" is rather general. The data used to support this statement were not shown and discussed until Section 3.1. We believe the related figures and descriptions in Section 3.1 are still very important. Some descriptions on DISCOVER-AQ data in the following paragraph were merged into Section 3.1 to avoid repetition.

Possessive is not necessary when describing data from a model or measurement platform. For example, replace "NUWRF's day time surface air temperature" with "NUWRF day time surface air temperature". There are many other instances where apostrophes are used, but aren't required.

Please identify and correct these.

We agree and modified the related sentences.

The equation used to infer isoprene emissions relies on OH concentration and boundary layer height as input. I would like to see some discussion and evaluation of the diurnal variability of these parameters, as these are used as input in Eq. (2) to estimate isoprene emissions from the isoprene concentration observations.

A rough evaluation of NAQFC OH is in Section 2.2.2, with the descriptions on its diurnal variability added: "Close to the estimated OH concentrations of approximately $2\text{-}6\times10^6$ molecule/$cm^3$ near Houston on 16 September, 2006 (Warneke et al., 2010), the simulated PBL OH on 11 September, 2013 range from $\sim1.8\times10^6$ to $\sim4.0\times10^6$ molecule/$cm^3$ along the P-3B around Conroe. The OH levels are higher in late morning and around noon ($>3.1\times10^6$ molecule/$cm^3$) than in the afternoon ($\sim1.8\times10^6$ molecule/$cm^3$), qualitatively consistent with the observations in downtown Houston in May 2009 (Czader et al., 2013). The averaged OH on all P-3B flight days is within the same range, following similar diurnal variability."

The evaluation of model PBLH is included in Section 3.1.1 and Table 3 for 11 September. The text briefly mentioned that the ctrl run PBLH "… are likely more realistic referring to the observed isoprene vertical profiles." The multi-day mean numbers are not significantly different: ~0.6/1.6/1.8km (AM/~Noon/PM).

**Specific Comments:**

Page 4, line 4: What emissions come from the atmosphere to the canopy? Do the authors mean the emissions consumed/deposited within the canopy (term $\rho$ in Eq. (1))? If this is not considered in the emission model used in this study, then please clarify that this value is set to 1 to avoid confusion.

We clarified that $\rho$ is assumed to be 1.0 in MEGAN v2.1.

Page 5, lines 15-18: What is the difference in land cover between the IGBP-derived land cover that is used in this study and the default used in MEGAN to justify using an updated land cover map? Some discussion of how using this updated land cover impacts isoprene emission modelling is needed.

The only two land use/land cover options currently available in WRF are USGS (24-category, Advanced very-high-resolution radiometer (AVHRR)-based) and IGBP-modified MODIS (20-category):

http://www2.mmm.ucar.edu/wrf/users/docs/user_guide_V3/users_guide_chap3.htm#_Land_Use_and

A third option is available in NUWRF called "University of Maryland", which is also AVHRR-based (Tao et al., 2013).

The MODIS option was chosen for this study because it was relatively more up-to-date than the other available options, and it is expected to lead to better meteorological output which was used to drive MEGAN. See the language "..which reflect more recent conditions than the other available options (Tao et al., 2013; Yu et al., 2012)."

None of these options represents the 2013 conditions. It is understood that many factors modified the US landscape such as biomass burning during $SEAC^4RS$ in some regions. We are currently in

the process of testing and comparing LIS/NUWRF simulations with surface type inputs of IGBP-modified MODIS and VIIRS (Visible Infrared Imaging Radiometer Suite) (Zhang, R. et al., 2016). See the figure below comparing these two inputs, which is also included in the SI.

[Figure]

In Section 3.3, we recommended to use up-to-date land use/land cover input data in both (NU)WRF and MEGAN in future studies. Considering MEGAN needs a different format of land cover input, this would require mapping recent satellite information to the Community Land Model PFT categories, as in some previous studies (e.g., Lawrence and Chase, 2007).

Zhang, R., Huang, C., Zhan, X., Dai, Q., and Song, K. (2016), Development and validation of the global surface type data product from S-NPP VIIRS, Remote Sens. Lett., 7, 51-60, doi: 10.1080/2150704X.2015.1101649.

Lawrence, P. J., and T. N. Chase (2007), Representing a new MODIS consistent land surface in the Community Land Model (CLM 3.0), J. Geophys. Res., 112, G01023, doi:10.1029/2006JG000168.

Page 5, lines 23-24: What is "the urban surface option"?
We meant to activate the WRF urban canopy model (Chen et al., 2011) for the Noah LSM. However, since NUWRF couples the land surface model from LIS in which this does not seem to be activated. Therefore, this line has been removed.

Page 5, line 26; Table 3 footnote: What is "full clocks"? This isn't standard terminology. Rather describe this in a way that can be understood.
Rewritten as: "..recorded hourly at 00:00 (minute:second)".

Page 7, lines 4-5: Why mention the August western US flights? Seems it has no relevance to this study and so can be removed.
This sentence was removed and the nearby sentences were reworded.

Page 11, line 32: Is "vastly similar" correct? In the next clause, the authors state that the difference is >30%.
"Vastly similar" refers to the comparisons among model runs. ">30%" refers to the model-observation differences. So it is correct. We modified this sentence slightly to avoid confusion.

Page 12, lines 29-31: Why does resolution induce a difference in isoprene emissions in the

atmospheric initialization sensitivity test in Figure 7c) and not in Figure 4c)?

[Figure]

[Figure]

This is a very good question. There can be three reasons for the discrepancies among 12 km, 4 km and aircraft-derived emissions. Two of them were already included in the earlier paragraph: 1) "…This is in part due to the coolest temperature from this NUWRF run, especially in the afternoon, as well as its weakest photosynthetically active radiation than the 12 km simulations' (i.e., by ~10 W/m$^2$ on average during the daytime)." 2) "This may also be resulting from some limitations of MEGAN's parameterization and uncertainty in its other inputs (e.g., PFT and LAI) on grid scale."

A third reason has just been identified and added, defined as the "representation error" due to discrepancies in different data resolutions as well as neglecting horizontal transport in deriving emissions from aircraft data. Multiple P-3B aircraft data points correspond to several NUWRF model grids, and the averaged NUWRF-MEGAN emissions were used in the comparisons. The bottom panels in the left figure (adapted from Figure 6b of the paper) show that for the grids collocated with the west side of the aircraft spiral (built-up land as indicated in the upper panel Google image), 4 km emissions are significantly lower than 12 km emissions. In the forestal/agricultural areas southwest to the spiral (highlighted by red arrows in the figure), 4 km and 12 km emissions are similar. It is very likely that higher emissions are from regions outside the spiral as the 4 km results indicate, and they were transported to the aircraft-sampled regions. On the 12 km grid, spatial variability of the emissions is not well distinguished, and the impact of horizontal transport did not appear to be evident. As a result, we see smaller discrepancies between the modeled and observation-derived.

Page 16, lines 1-2: Please provide references to back up the statement that many model comparison studies don't adequately assess the impact of model inputs.

A sentence in the introduction also supports this point. A few references have been provided there: "e.g., Millet et al., 2008; Warneke et al., 2010; Canty et al., 2015; Hogrefe et al., 2011". The Carlton and Baker (2011) includes some discussions on the impact of meteorological input. We added "cited in Section 1" to this sentence.

Figure 3: Please increase the size of the points in the Figure 3a) Observations panel so that the reader can easily compare the observations and model or instead show a scatterplot of the model versus the observations and include regression statistics.

The original observational data were regridded to the 12 km model grid for this plot, as mentioned in the figure caption, and therefore the data shown were in grid-size. We agree that this panel was a bit hard to read and regridding is unnecessary for making this plot. We now just show the original observations at their actual locations, in bigger size. Bulk statistics would not be quite relevant to

the point we made from this figure.

Figure 3 caption: Please say where the temperature observations are from. Are these NCEP?
Added.
We also added in the caption that "Note that for (b) and (c), warm (cool) colors indicate high (low) soil moisture values, opposite to the commonly used color scheme in hydrological studies." In most hydrological studies, warmer colors indicate drier conditions. We flipped the color scheme so that it is consistent with the usage in emission and air quality studies.

---

## Author Response (AR2)

**Response to the Editor's and Referee #1's comments**

Editor's comments to the Author: The review of the revised manuscript has been generally positive with the exception of one more serious concern. I agree with the reviewer that this issue could be addressed relatively easily, or at least should be explained why the recommendation has not been taken up, in the response to this second review and in the manuscript is apropriate. For this reason I have also agreed with the reviewer to ask for a minor revision. Once the revisions have been submitted they will be reviewed by the topical editor (i.e., me) so no extended revision period will arise from this. Otherwise the manuscript is ready for publication. Many Thanks to the reviewers for their continued support and to the authors for all their efforst.

We thank the Editor's efforts on this manuscript and the second review by Referee #1. All of the Referee's major and minor comments have been addressed. Please see below our point-by-point response (in blue) to the comments (in black).

My only substantive comments have to do with Table 2 and the discussion around PBLH. It is not clear to me why no attempt was made to compare the modeled PBLH to the DIAL-HSRL data, given the authors have access to these data and show them in Fig. 2a. They already extract other variables along the flight track, so also extract PBLH and providing some quantitative evaluation against the measurements seems both feasible and necessary.

We appreciate the careful re-review and have made changes to the paper accordingly.

The main reason for not presenting a quantitative evaluation of the modeled PBLH was that the DIAL-HSRL dataset comes with different temporal resolution (10s) and data completeness, and these mixed layer heights may be highly uncertain.

The mean and standard deviation of DIAL-HSRL data near the "isoprene volcano" are now included in Table 2. To ensure the readers correctly understand the comparisons, we also added a note below the table: "Compared with the other used DC-8 datasets, the DIAL-HSRL dataset has different temporal resolution and completeness. Its mixed layer heights may be highly uncertain." In Section 3.1.1, we are also careful with the use of language to introduce the model/DIAL-HSRL discrepancies: "They may be closer to the reality referring to the DIAL-HSRL data in Figure 2a, which can also be uncertain."

Along these lines, the Table 2 & 3 captions are not correct (with current table contents), as they state they show the 'air temperature and PBLH performance' – but in reality they show air temperature performance and modeled PBLH values (not performance as it has not be evaluated). Of course, if this is changed the include PBLH comparison metrics against the observations, this point is no longer relevant.

Thanks for pointing this out. The revised Table 2 caption is "NUWRF PBLH and air temperature performance along the DC-8 flight path". The revised Table 3 caption is "NUWRF-simulated median PBLH (km) and daytime (6-17 local standard time) surface air temperature performance"

Finally, it is not clear why Table 2 does not show the surface air temperature mean bias like Table 3 does. I do not see any reason for these two tables to use different temperature metrics.

The evaluation for Missouri and Texas regions slightly differs, with the former more focused on spatial variability at ~13 local standard time and the latter more focused on the daytime time series at Conroe. We do agree that the similar set of evaluation metrics should be shown. Figure S3 has

been added to show the spatial variability of air temperature biases near 13 local standard time of this day. Also, in Table 2, we now summarize mean and standard deviation values of air temperature and PBLH for all observational and modeling datasets, from which the mean biases can be estimated. Related discussions were extended in Section 3.1.1.

Minor comments:

P6 L20-21: Add the reference provided in the response to reviewers
Done.

P7 L3: Rate constant should be in scientific notation: 1.01x10^-10
Done.

P7 L5: after qualitatively evaluated --> after qualitative evaluation OR after being qualitatively evaluated
Changed to: "after being qualitatively evaluated"

P7 L20: accuracy of +/-5% -- is there a reference for this?
"+/-5%" was found in the aircraft observation data file. The source of the observation data is included in Section 5 the manuscript.

P7 L30: north to --> north of
Done.

P7 L34: located in downtown Houston area --> located in downtown Houston OR located in the downtown Houston area
Changed to: "located in the downtown Houston area"

P8 L1: suggest replacing biogenic with isoprene here, as monoterpene emissions are not necessarily at a miminum overnight
"Isoprene" has been added.

P8 L15: in the eastern Texas --> in eastern Texas
Done.

P8 L17: missing units for predicted OH value on 11 Sep
Added.

P8 L18 of the close magnitudes --> close in magnitude
Done.

P8 L21: range from --> ranged from
Done.

P8 L22: along the P-3B -- missing a word; perhaps along the P-3B flight track?
Added "flight tracks"

P10 L3: on a specific day of 11 September --> on the specific day of 11 September, or better yet just on 11 September
Removed "a specific day of".

P10 L10: at around the Conroe site --> around the Conroe site
Done.

P10 L20: same comment as before about changing biogenic to isoprene (or biogenic isoprene)
Modified.

P11 L22: significantly changes --> significant changes
Done.

P12 L28 & elsewhere: daytime-integrated emissions are mentioned but never provided. A table with these values would make the text in this section easier to follow.
Good suggestion. These numbers are now included in Figure 7.

P14 L22: radiation product --> radiation products
Done.

P15 L13: could the 10-20% overestimate of the observation-derived estimates explain some of the previously described underestimate of the 4-km run? This would be worth noting.
We believe the main reasons for the low 4 km NUWRF-MEGAN emissions are "This is possibly resulting from some limitations of MEGAN's parameterization, uncertainty in its inputs on small scale, as well as the representation error and neglecting horizontal transport in deriving emissions from aircraft data.", as mentioned in the abstract and the text. The biases in observed isoprene due to non-biogenic sources could only affect the observation derived emissions.

P15 L20: of the similar magnitude --> of similar magnitude
Done.

P16 L6: Suggest starting a new paragraph at 'This study emphasizes the importance…'
Done.

P16 L14: variability of the atmospheric composition --> variability of atmospheric composition
Done.

P34 Fig. 8 caption: Specify that the model results are from the 12 km NUWRF ctrl run. Also comparing with should be compared with.
Done.

P35 Table 1 caption: descriptions on --> descriptions of
Done.

[revised manuscript text omitted]